# Rtt105 regulates RPA function by configurationally stapling the flexible domains

Sahiti Kuppa[1], Jaigeeth Deveryshetty[1,10], Rahul Chadda[1,10], Jenna R. Mattice[2,10], Nilisha Pokhrel[3,9], Vikas Kaushik[1], Angela Patterson[2], Nalini Dhingra [4], Sushil Pangeni[5], Marisa K. Sadauskas[1], Sajad Shiekh [6], Hamza Balci [6], Taekjip Ha [5,7,8], Xiaolan Zhao [4], Brian Bothner[2] & Edwin Antony [1,3] ✉

Replication Protein A (RPA) is a heterotrimeric complex that binds to single-stranded DNA (ssDNA) and recruits over three dozen RPA-interacting proteins to coordinate multiple aspects of DNA metabolism including DNA replication, repair, and recombination. Rtt105 is a molecular chaperone that regulates nuclear localization of RPA. Here, we show that Rtt105 binds to multiple DNA binding and protein-interaction domains of RPA and configurationally staples the complex. In the absence of ssDNA, Rtt105 inhibits RPA binding to Rad52, thus preventing spurious binding to RPA-interacting proteins. When ssDNA is available, Rtt105 promotes formation of high-density RPA nucleoprotein filaments and dissociates during this process. Free Rtt105 further stabilizes the RPA-ssDNA filaments by inhibiting the facilitated exchange activity of RPA. Collectively, our data suggest that Rtt105 sequesters free RPA in the nucleus to prevent untimely binding to RPA-interacting proteins, while stabilizing RPA-ssDNA filaments at DNA lesion sites.

Replication protein A (RPA) is an essential single-stranded DNA (ssDNA) binding protein that coordinates almost all aspects of DNA metabolism including DNA replication, repair, and recombination[1]. The myriad functions of RPA are facilitated through high affinity interactions with ssDNA ($K_D < 10^{-10}$M) and physical interactions with over three dozen proteins[2]. With respect to the functions of RPA in double strand DNA break repair, current models posit that recognition of ssDNA by RPA triggers the DNA damage response (DDR). Coating of RPA on ssDNA serves as a nucleoprotein hub for the recruitment of several proteins including kinases that trigger downstream DNA repair pathways[3]. For example, yeast Mec1 or human ATM/ATR kinases are recruited on to the RPA-ssDNA substrate[4]. How such protein-protein interactions with RPA are prevented in the absence of ssDNA remains a mystery.

Functional specificity for the myriad DNA metabolic functions of RPA is facilitated by a complex series of oligosaccharide/oligonucleotide binding (OB) folds/domains that are distributed across a heterotrimeric structural complex made of RPA70, RPA32 and RPA14 subunits (Fig. 1a)[5,6]. OB-domains A, B, C, and F are in RPA70[7]. OB-domain D and a winged-helix (wh) domain are situated in RPA32, and RPA14 houses OB-domain E. OB-F and the wh-domain are primarily responsible for mediating protein-protein interactions and are thus termed Protein-Interaction-Domains (PID[70N] and PID[32C], respectively)[8,9]. OB-A, B, C and D contribute most to ssDNA binding and are termed DNA-Binding-

[1]Department of Biochemistry and Molecular Biology, Saint Louis University School of Medicine, St. Louis, MO 63104, USA. [2]Department of Chemistry and Biochemistry, Montana State University, Bozeman, MT 59717, USA. [3]Department of Biological Sciences, Marquette University, Milwaukee, WI 53201, USA. [4]Molecular Biology Department, Memorial Sloan Kettering Cancer Center, New York, NY 10065, USA. [5]Department of Biophysics, Johns Hopkins University, Baltimore, MD 21218, USA. [6]Department of Physics, Kent State University, Kent, OH 44242, USA. [7]Department of Biophysics and Biophysical Chemistry, Johns Hopkins University, Baltimore, MD 21205, USA. [8]Howard Hughes Medical Institute, Baltimore, MD 21205, USA. [9]Present address: Laronde Bio, Cambridge, MA, USA. [10]These authors contributed equally: Jaigeeth Deveryshetty, Rahul Chadda, Jenna Mattice. ✉e-mail: edwin.antony@health.slu.edu

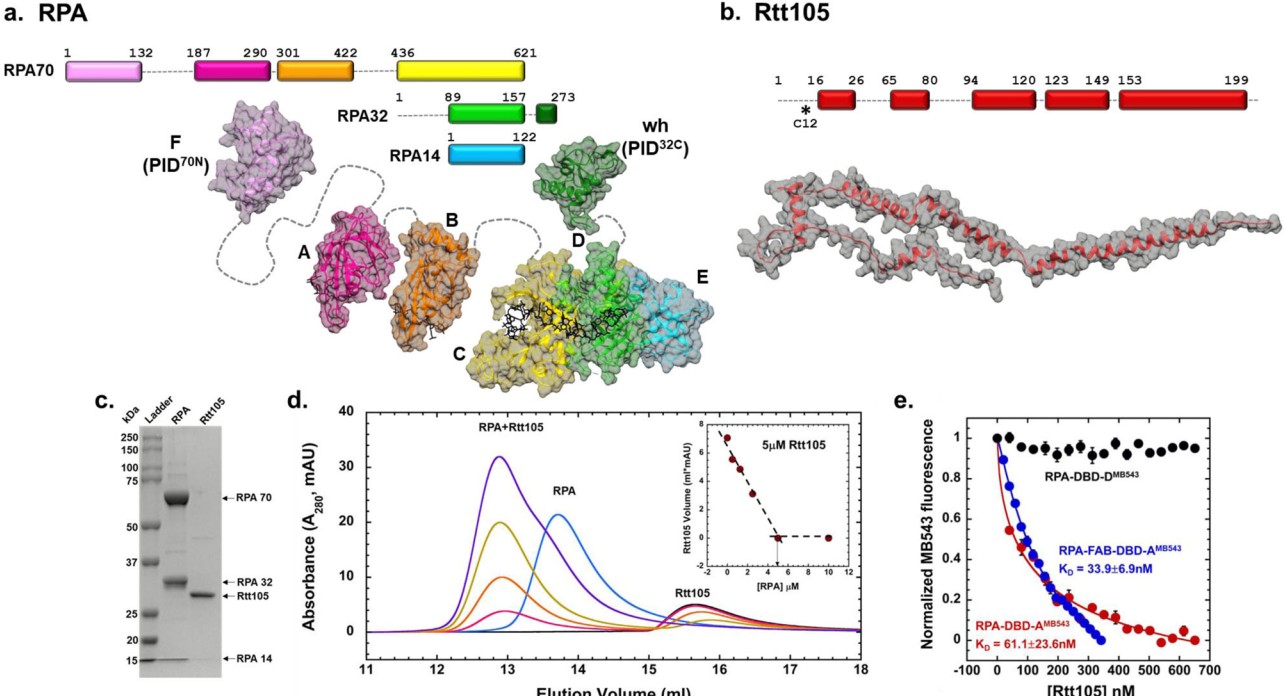

**Fig. 1 | Rtt105 forms a stoichiometric complex with RPA. a** Schematic of the OB-domains of the three *S. cerevisiae* RPA subunits is depicted along with the structures of the individual domains. The domains are spread apart for clarity and the dotted lines denote the flexible linkers connecting the domains. ssDNA bound to the individual DBDs (A, B, C and D) is shown as sticks. DBDs-C, D and E are from the cryoEM structure of *S. cerevisiae* RPA (PDB: 6I52). Homology models for DBD-F (PDB: 5N8A), DBD-A & DBD-B (PDB: 1JMC), and winged helix domains (PDB:4OU0) were built with Swiss-Model using the listed PDB files as template. DNA from 1JMC is docked onto DBD-A and B homology models. **b** AlphaFold (AF-P40063-F1) model of Rtt105 is shown with predicted α-helices. *Denotes Cys-12 used to generate fluorescently labeled Rtt105. **c** SDS-PAGE analysis of recombinantly purified RPA and Rtt105. **d** Size exclusion chromatographic (SEC) analysis of Rtt105 (5 μM) and RPA (0–5 μM) show RPA, Rtt105, and the Rtt105-RPA complex migrating as defined species. Insert shows the change in the Rtt105 peak volume as a function of RPA concentration and stoichiometric complex formation between Rtt105 and RPA. **e** Changes in RPA-DBD-A$^{MB543}$ or RPA-DBD-D$^{MB543}$ fluorescence was measured after addition of increasing concentrations of Rtt105. Rtt105 binding does not influence the fluorescence of RPA-DBD-D$^{MB543}$ but quenches RPA-DBD-A$^{MB543}$ fluorescence. F-A-B labeled at DBD-A$^{MB543}$ binding to Rtt105 reveals a higher binding affinity compared to full-length RPA. Data were fit as described in the Methods to obtain $K_D$ values. Mean values +/− SE from *n* = 3 biologically independent experiments are shown.

Domains (DBDs). The roles of OB-F and OB-E in binding ssDNA are uncertain. The DBDs and PIDs are connected by flexible linkers of varying lengths and thus can adopt a wide variety of structural assemblies on and off the DNA[2], collectively termed 'configurational arrangements' in RPA. Conformational changes upon ssDNA binding and protein interactions are also quite extensive within the individual domains[10].

In response to cellular DNA metabolic needs, RPA is shuttled from the cytoplasm into the nucleus. In *Saccharomyces cerevisiae*, the nuclear localization of RPA is facilitated by a chaperone-like protein called Regulator of Ty1 Transposition 105 (Rtt105)[11]. In higher eukaryotes, RPA-interacting protein (RPAIN or RIPα) appears to be the functional homolog. *RTT105* was originally discovered in a genome-wide screen as interacting with genes associated with genome maintenance pathways[12]. Rtt105 physically interacts with RPA in the cytoplasm and mediates nuclear localization in complex with Kap95, a key karyopherin-beta protein that belongs to the importin-exportin family of nuclear transport protein complexes[11]. Consequently, deletion of Rtt105 or mutations that perturb the RPA-Rtt105 interaction give rise to defects in cell growth, DNA repair, and recombination[13,14].

Rtt105 is a 24 kDa protein (Fig. 1b) and proposed to stabilize an extended conformation of RPA where all the OB-domains are stretched out[11,14,15]. In addition to the nuclear transport functions, recent studies propose Rtt105 to enhance the ssDNA binding properties of RPA[11,14]. Depending on the number of DBDs bound to ssDNA, RPA can adopt distinct DNA binding modes where varying number of nucleotides are occluded by the RPA complex[16]. The DBDs of RPA are also dynamic in

nature and thus, while the protein is macroscopically bound to the ssDNA, each domain can exist in microscopically unbound states[17–19]. We recently showed that RPA can be envisioned to interact with ssDNA as two functional halves: a dynamic half consisting of the OB-F, DBD-A and DBD-B domains, and a less-dynamic half with DBD-C, DBD-D and OB-E (trimerization core or Tri-C) contributing more to the stability of RPA-ssDNA interactions[20].

RPA binds to ssDNA with very high affinity ($K_D < 10^{-10}$M)[17,21]. Yet, a substantial increase in RPA binding affinity to ssDNA was shown in the presence of Rtt105 which led to a model where Rtt105 binds and promotes a configuration of RPA that extends out its DBDs such that maximal DNA binding contacts are promoted[11,14,15]. Rtt105 inherently does not possess DNA binding activity and RPA intrinsically binds ssDNA with very high affinity. More strikingly, Rtt105 is not bound to the RPA-ssDNA complex. Thus, the finding that Rtt105 enhances the DNA binding activity of RPA, while not part of the DNA-RPA complex is perplexing, given that experimental tools such as electrophoretic mobility shift analysis (EMSA) do not have the required resolution to detect changes in sub-nanomolar biomolecular interactions. Furthermore, the rationale for Rtt105 stabilizing an extended conformation of RPA whilst having to transport it across the nuclear pore is also puzzling. To address these disparities, we undertook a detailed structural and mechanistic investigation of the Rtt105-RPA complex.

Here, we show that Rtt105 contacts multiple regions in RPA and conformationally compacts the multiple DBDs and PIDs through a configurational stapling mechanism. Shorter ssDNA substrates (~15–35 nt) engage the Rtt105-RPA complex and promote remodeling of Rtt105, and a transition-state Rtt105-RPA-ssDNA complex is

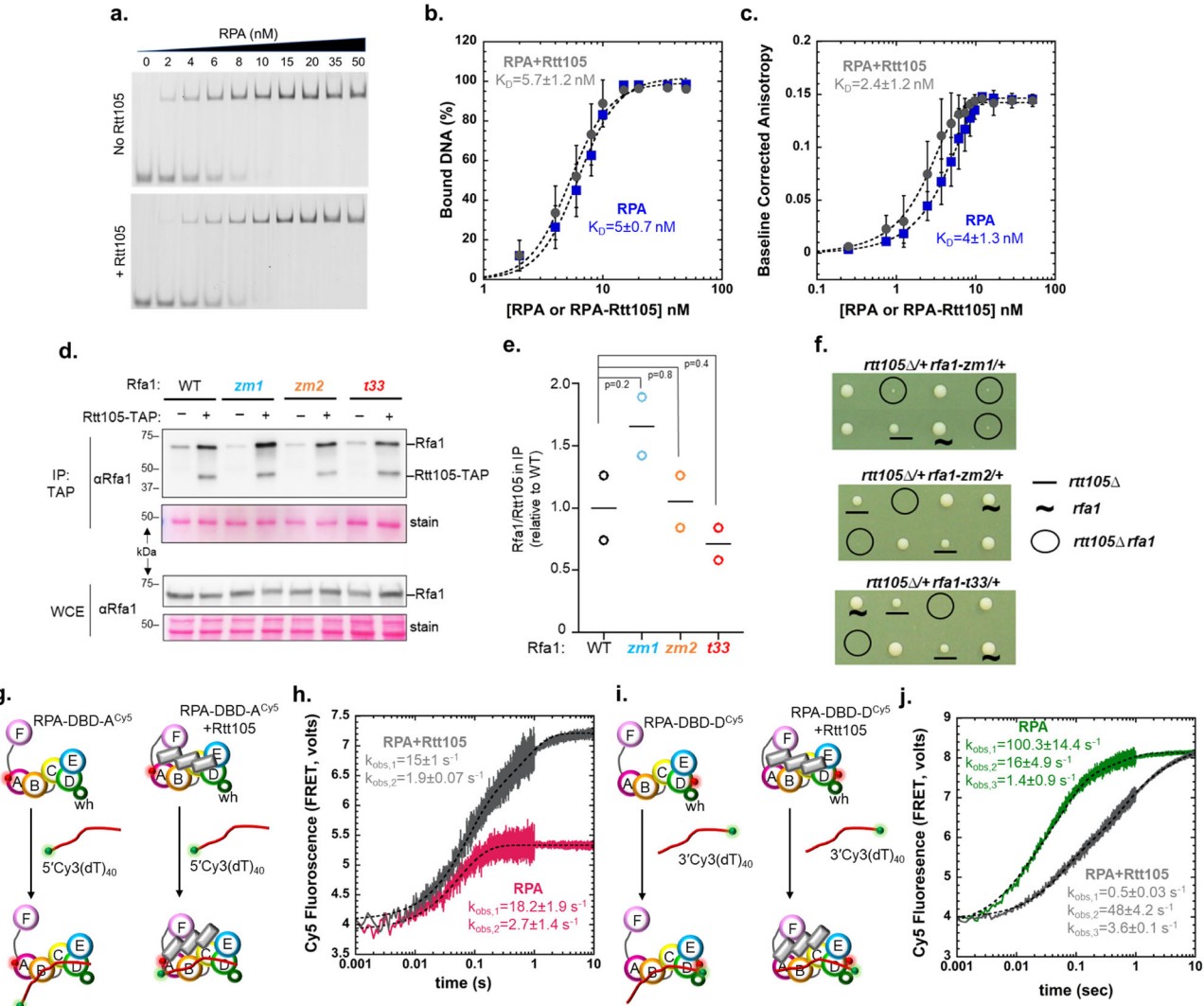

**Fig. 2 | Rtt105 alters the configuration of DBD-A on ssDNA and kinetics of DBD-D binding. a** EMSA of RPA binding to 5′-Cy5-(dT)$_{30}$ ssDNA and **b** quantitation shows no appreciable difference in DNA binding affinity in the absence ($K_D = 5 \pm 0.7$ nM) or presence of Rtt105 ($K_D = 5.7 \pm 1.2$ nM). **c** Fluorescence anisotropy measurements of RPA binding to 3′-FAM-(dT)$_{35}$ ssDNA shows no appreciable difference in DNA binding affinity in the absence ($K_D = 4 \pm 1.3$ nM) or presence of Rtt105 ($K_D = 2.4 \pm 1.2$ nM). **d** Rtt105 association with RPA is unaffected by its ssDNA binding ability. TAP-tagged Rtt105 was immunoprecipitated and RPA was detected with an anti-Rfa1 antibody. Representative immunoblots examining elute from the eluate (IP) and the whole cell extract (WCE) are shown. **e** Enrichment of Rfa1 present in the IP eluate relative to Rtt105 and normalized to wild type (WT). Mean and SEMs from two biological duplicates are plotted. *P* value obtained from Students 2-tailed *t*-test show that the differences in relative Rfa1 enrichments are not significant. **f** *rtt105Δ* sensitizes *rfa1* mutants with impaired ssDNA association. Representative tetrads of diploids heterozygous for indicated mutations are shown. **g, h** Stopped flow

fluorescence measurements of DNA binding were monitored by FRET-induced enhancement of Cy5 fluorescence upon Cy3 excitation. FRET pairs positioned on the 5′-DNA end (5′-Cy3-(dT)$_{40}$) and DBD-A of RPA (RPA-DBD-A$^{Cy5}$) show similar rates of DNA-induced changes +/− Rtt105 ($k_{obs,1} = 18.2 \pm 1.9$ s$^{-1}$, $15 \pm 1$ s$^{-1}$; $k_{obs,2} = 2.7 \pm 1.4$ s$^{-1}$, $1.9 \pm 0.07$ s$^{-1}$; in the absence and presence of Rtt105, respectively), but Rtt105 markedly enhances the amplitude of Cy5 fluorescence. **i, j** Similar experiments performed with FRET pairs positioned on the 3′-DNA end (3′-Cy3-(dT)$_{40}$) and DBD-D of RPA (RPA-DBD-D$^{Cy5}$) show an Rtt105 induced reduction in the rate of DNA binding ($k_{obs,1} = 100.3 \pm 14.4$ s$^{-1}$, $0.5 \pm 0.03$ s$^{-1}$; $k_{obs,2} = 16 \pm 4.9$, $48 \pm 4.2$ s$^{-1}$; $k_{obs,3} = 1.4 \pm 0.9$, $3.6 \pm 0.1$ s$^{-1}$in the absence and presence of Rtt105, respectively). But, unlike in **d**, both traces show similar changes in fluorescence amplitude. Representative stopped flow data averaged from seven to eight shots from one experiment are shown. For **b, c, h,** and **j**, mean values +/− SE from $n = 3$ biologically independent experiments are denoted.

observed. Longer ssDNA substrates (>35 nt) promote the binding of multiple RPA molecules and lead to dissociation of Rtt105. Interestingly, Rtt105 promotes formation of high-density RPA nucleoprotein filaments on ssDNA. In the absence of ssDNA, we find that Rtt105 blocks interactions between RPA and RPA-interacting proteins (RIPs) such as the homologous recombination mediator Rad52. ssDNA binding to Rtt105-RPA remodels the Rtt105-RPA-ssDNA complex and promotes RIP engagement. Posttranslational modifications of RPA further contribute to the remodeling of the RPA-Rtt105 complex. Finally, Rtt105 blocks the facilitated exchange activity of RPA thereby contributing to the stability of RPA nucleoprotein filaments. Thus, we

here present new functional roles for Rtt105 where it sequesters RPA in the nucleus, drives formation of densely packed RPA filaments, and serves as a negative regulator by blocking spurious interactions with RPA-interacting proteins in the absence of ssDNA.

## Results

### Rtt105 forms a stoichiometric complex with RPA
Rtt105 physically interacts with RPA and was shown to copurify as a complex[11]. To obtain the stoichiometry of the complex, using recombinantly purified proteins (Fig. 1c) we analyzed formation of the Rtt105-RPA complex as a function of RPA concentration using size exclusion

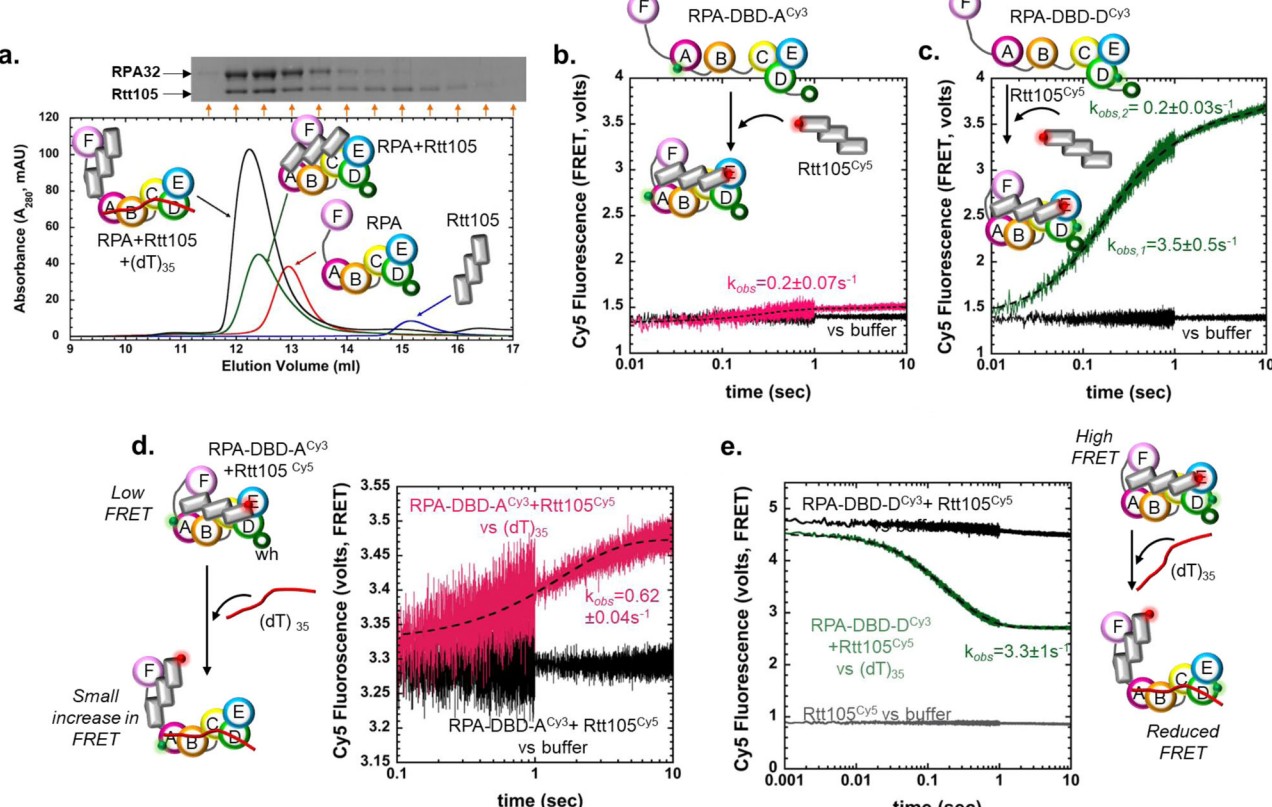

**Fig. 3 | Rtt105 forms a higher order complex with RPA bound to short ssDNA substrates. a** Size exclusion chromatography (SEC) analysis of Rtt105, RPA, and Rtt105·RPA show complex formation between the proteins. Addition of equimolar amounts of $(dT)_{35}$ drive formation of a higher order Rtt105-RPA-$(dT)_{35}$ complex. SDS page analysis of the SEC fractions (insert) shows RPA and Rtt105 comigrating as a complex in the presence of DNA. A small fraction of free Rtt105 is also observed. **b** The kinetics of RPA binding to Rtt105 was measured using FRET with Cy5-labeled Rtt105 and either RPA-DBD-A$^{Cy3}$ or **c** RPA-DBD-D$^{Cy3}$. Cy3 positioned on DBD-D produces a larger FRET-induced Cy5 enhancement compared to DBD-A suggesting that the N-terminal region of Rtt105 resides closer to DBD-D of RPA. The data for RPA-DBD-A in **b** fits well to a single exponential model ($k_{obs} = 0.2 \pm 0.07$ s$^{-1}$) whereas the RPA-DBD-D data is better described by a double exponential ($k_{obs,1} =$

$3.5 \pm 0.5$ s$^{-1}$, $k_{obs,2} = 0.2 \pm 0.03$ s$^{-1}$). These data suggest faster, initial engagement of Rtt105 towards the trimerization core (Tri-C) of RPA. Stopped flow experiments show transition from either a preformed **d** RPA-DBD-A$^{Cy3}$:Rtt105$^{Cy5}$ or **e** RPA-DBD-D$^{Cy3}$:Rtt105$^{Cy5}$ complex to remodeled Rtt105$^{Cy5}$-RPA$^{Cy3}$-$(dT)_{35}$complexes upon addition of ssDNA. A small increase in Cy5 fluorescence is observed for the RPA-DBD-A$^{Cy3}$:Rtt105$^{Cy5}$ complex. In contrast, a sharp decrease in Cy5 signal is observed for the RPA-DBD-D$^{Cy3}$:Rtt105$^{Cy5}$ complex. The Cy5 fluorescence does not reach the baseline signal for Rtt105$^{Cy5}$. The remodeling data fits well to a single step model ($k_{obs} = 3.3 \pm 0.96$ s$^{-1}$). Representative stopped flow data averaged from seven to eight shots from one experiment are shown. Mean values +/− SE of the observed rates from $n = 3$ biologically independent experiments are denoted.

chromatography (SEC). We observe concentration-dependent stoichiometric binding between RPA and Rtt105 (Fig. 1d). To determine a $K_D$ value for the interaction, we followed the change in fluorescence in RPA upon complex formation with Rtt105. We used fluorescent versions of RPA where either DBD-A or DBD-D carry a site-specific environment-sensitive MB543 fluorophore[17,22,23]. Upon binding to Rtt105 we observe a change in fluorescence for RPA-DBD-A$^{MB543}$, but not for RPA-DBD-D$^{MB543}$ (Fig. 1e). Rtt105 binds to RPA with high affinity ($K_D = 61.1 \pm 23.6$ nM; Fig. 1e) and the selective change in RPA-DBD-A$^{MB543}$ fluorescence suggests that at least a part of Rtt105 is situated/bound in proximity to DBD-A. Furthermore, an F-A-B version of RPA containing just OB-F, DBD-A and DBD-B interacts with Rtt105 (Supplementary Fig. 1a). Thus, the region around OB-F and DBD-A likely makes physical contacts with Rtt105. The F-A-B domain binds to Rtt105 with higher affinity ($K_D = 33.9 \pm 6.9$ nM; Fig. 1e). These data suggest that one or more of the Rtt105 binding sites in the F-A-B region is occluded in the context of the full-length RPA complex. A recent study suggested that a Val-106 to Ala substitution in OB-F (RPA70 subunit) was sufficient to abolish interaction with Rtt105[14]. However, in our analysis, this single point substitution in RPA (RPA$^{V106A}$) is not sufficient to perturb the interaction (Supplementary Fig. 1b) suggesting that additional contacts must exist between Rtt105 and RPA.

## Rtt105 alters the ssDNA bound configurations of the DNA binding domains (DBDs) of RPA

Recent studies have shown that Rtt105 enhances the ssDNA binding activity of RPA[11]. More perplexingly, Rtt105 is not bound to the RPA-ssDNA complex in pull-down experiments. Thus, how Rtt105 modulates the DNA binding activity of RPA, while not part of the RPA-ssDNA complex, is unresolved. In both these earlier studies, control experiments performed in the absence of Rtt105 showed only ~50% of RPA bound to ssDNA oligonucleotides at equimolar concentrations[11,14]. These data are inconsistent with the high-affinity and stoichiometric DNA binding behavior of RPA[17,22]. Thus, we first repeated these experiments, under similar experimental conditions, and do not observe this behavior. RPA binds stoichiometrically to a 5′-Cy5-$(dT)_{30}$ oligonucleotide in electrophoretic mobility band-shift analysis (EMSA) and a preformed Rtt105-RPA complex does not influence the ssDNA binding activity of RPA ($K_D = 5 \pm 0.7$ and $5.7 \pm 1.2$ nM for RPA and RPA-Rtt105, respectively; Fig. 2a, b). Since EMSA experiments may not have the resolution to tease apart subtle differences in high-affinity biomolecular interactions, we used fluorescence anisotropy to further quantitate binding. RPA and the RPA-Rtt105 complex bound stoichiometrically to a 5′-FAM-$(dT)_{35}$ ssDNA substrate and we do not see a measurable Rtt105-induced

enhancement of RPA-ssDNA binding activity ($K_D = 4 \pm 1.3$ and $2.4 \pm 1.2\,nM$ for RPA and RPA-Rtt105, respectively; Fig. 2c). Nevertheless, in both these experiments, the high affinity stoichiometric interactions between RPA and ssDNA make it experimentally difficult to decipher a role for Rtt105 in modulating the overall binding affinity of RPA to ssDNA.

These results suggest that to a large degree, RPA association with Rtt105 is a biochemical feature that can be functionally distinct from its ssDNA binding activity. This notion of two separable features of RPA is supported by our in vivo data, where we tested the cellular effects of deletion of Rtt105 along with three distinct DNA binding mutants of RPA. zm1 (K494A) and zm2 (N492D, K494R, K494R) mutants of RPA carry amino acid substitutions close to the $Zn^{2+}$-finger binding domains in DBD-C and have reduced ssDNA binding activity[24]. t33 is a well characterized RPA mutant with a S373P substitution in DBD-B[25,26]. Immunoprecipitation of Rtt105 from cells expressing these mutant variants of RPA show that the physical interaction between the two proteins is not perturbed by these mutations in RPA (Fig. 2d, e). From a genetic point view, the lack of measurable influence on global RPA-DNA binding by Rtt105 (Fig. 2b, c) predicts that mutants impairing RPA binding to ssDNA in cells lacking Rtt105 should have additive genetic interactions. Indeed, we find that RPA mutations that displayed reduced ssDNA binding activities were additive when combined with *rtt105* null cells when assayed for growth (Fig. 2f). These in vivo data support the idea that RPA-Rtt105 interactions and RPA-ssDNA binding are two separable functions.

## Rtt105 influences the ssDNA binding kinetics and configurations of individual DBDs of RPA

Since RPA has four DBDs, its interaction with ssDNA can be defined in terms of macroscopic binding and microscopic dynamics of DBDs[17,18]. Each DBD has intrinsic on/off ssDNA binding properties that describe the kinetics of their interactions. Thus, at any given point, RPA as a complex can remain stably bound to ssDNA while one or more DBDs can be remodeled or displaced[2]. Such dynamic interactions allow RPA to interact with other proteins while bound to ssDNA[10]. Since the influence of Rtt105 on the equilibrium ssDNA binding properties of RPA cannot be reliably measured because of the macroscopic binding interactions, we next tested whether the kinetics of RPA-ssDNA interactions (especially the DBDs) were influenced by Rtt105. We used fluorescent RPA carrying Cy5 on either DBD-A or DBD-D and captured the kinetics of binding to $(dT)_{40}$ oligonucleotides labeled with Cy3 at either the 5′ or 3′ end. Changes in Förster resonance energy transfer (FRET) induced Cy5 fluorescence were monitored by exciting Cy3 in a stopped flow fluorometer (Fig. 2g–j). Since the DBDs of RPA bind ssDNA with defined polarity[27,28], with DBD-A and DBD-D residing close to the 5′- and 3′ ends, respectively, the change in fluorescence in each experiment reflects the configurational changes of the respective DBDs with respect to their cognate DNA termini positions[17]. These data show that DBD-A binds with ~similar rates to the 5′ end-labeled DNA in the absence or presence of Rtt105 (Fig. 2g, h). However, the fluorescence signal amplitude in the presence of Rtt105 is twice that observed for RPA alone (Fig. 2h). In contrast, Rtt105 reduces the rate of DBD-D binding to the 3′ end-labeled DNA (Fig. 2i, j), while the signal amplitudes remain similar (Fig. 2j). These data suggest that Rtt105 differentially influences the two DBDs of RPA and support a model where Rtt105 could drive formation of specific configurations RPA (described below).

## Rtt105-RPA-ssDNA complexes are detected on short DNA substrates

The experiments above show that Rtt105 influences the configurations of the DBDs of RPA but does not reveal whether Rtt105 remains in complex with RPA in the presence of ssDNA. To directly test formation of the Rtt105-RPA-ssDNA complex we performed SEC analysis of the

individual proteins and their complexes. Rtt105 (24 kDa) and RPA (114 kDa) migrate as distinct species due to large differences in their molecular weight and migrate together as a complex when pre-mixed (Fig. 3a). However, when equimolar amounts of $(dT)_{35}$ are added to the complex, Rtt105 primarily remains bound as a higher order Rtt105-RPA-$(dT)_{35}$ complex. A small fraction of free Rtt105 and RPA-$(dT)_{35}$ complexes are also observed (Fig. 3a and Supplementary Fig. 1c, d). These data show that the configurational changes in RPA observed in the stopped flow experiments (Fig. 2) are induced while both Rtt105 and DNA are simultaneously bound to RPA. We propose that the RPA-Rtt105 complex binds ssDNA and both proteins are likely reconfigured as the DBDs engage onto ssDNA. Thus, the interactions between RPA and Rtt105 are different in the DNA bound versus unbound states. Based on the changes in observed fluorescence between the ends of the DNA and the DBDs, and the higher Rtt105 binding affinity for the F-A-B half of RPA, we propose that upon ssDNA binding Rtt105 interactions with RPA are likely shifted towards the F-A-B part of RPA. This model suggests an active repositioning of Rtt105 on RPA upon ssDNA binding.

To experimentally capture this process, we generated fluorescently labeled Rtt105 to perform FRET analysis of the RPA-Rtt105 complex. Rtt105 has two non-conserved native Cys residues situated adjacently (Cys12 and Cys13, Supplementary Fig. 2a). Substitution of either or both Cys residues to Ser does not change the secondary structure, RPA binding, or RPA remodeling activity of Rtt105 (Supplementary Fig. 2). Thus, we generated a single Cys version of Rtt105 by converting Cys13 to Ser and fluorescently labeled Rtt105$^{C13S}$ with Cy5 using maleimide chemistry (Supplementary Fig. 2e, f). Generation of a functional fluorescent Rtt105 enabled us to investigate the assembly, remodeling, and disassembly of the Rtt105-RPA complex using stopped flow FRET. First, we measured the kinetics of Rtt105$^{Cy5}$ binding to RPA-DBD-A$^{Cy3}$ or RPA-DBD-D$^{Cy3}$ by exciting Cy3 and monitoring the changes in Cy5 fluorescence. A small increase in FRET-induced Cy5 fluorescence is observed when Rtt105$^{Cy5}$ binds to RPA-DBD-A$^{Cy3}$ (Fig. 3b), but a robust signal change is observed for RPA-DBD-D$^{Cy3}$ (Fig. 3c). These data suggest that in the absence of DNA, the N-terminal region of Rtt105 (where the Cy5 is positioned) is situated closer to DBD-D. The data for DBD-A$^{Cy3}$ and Rtt105$^{Cy5}$ binding fits to a single-step model ($k_{obs} = 0.2 \pm 0.07\,s^{-1}$; Fig. 3b) whereas the DBD-D$^{Cy3}$ and Rtt105$^{Cy5}$ data fits better to a two-step model ($k_{obs,1} = 3.5 \pm 0.5\,s^{-1}$, $k_{obs,2} = 0.2 \pm 0.03\,s^{-1}$; Fig. 3c). Conservative interpretation of the kinetics suggests faster initial binding of the N-terminal region of Rtt105 closer to DBD-D followed by slower binding/rearrangements to other regions of RPA.

Next, to obtain the kinetics of ssDNA-induced remodeling of the Rtt105-RPA complex, we preformed either the low-FRET RPA-DBD-A$^{Cy3}$:Rtt105$^{Cy5}$ or high-FRET RPA-DBD-D$^{Cy3}$:Rtt105$^{Cy5}$ or complex and monitored the change in Cy5 fluorescence upon addition of $(dT)_{35}$ ssDNA. When the preformed RPA-DBD-A$^{Cy3}$:Rtt105$^{Cy5}$ complex is mixed with $(dT)_{35}$, a small enhancement in Cy5 fluorescence is observed ($k_{obs} = 0.62 \pm 0.04\,s^{-1}$, Fig. 3d). In the corollary experiment, a rapid, single-step transition from the high-FRET to low-FRET state is observed when the RPA-DBD-D$^{Cy3}$:Rtt105$^{Cy5}$ complex is mixed with $(dT)_{35}$ ($k_{obs} = 3.3 \pm 1\,s^{-1}$; Fig. 3e). However, the fluorescence signal does not decrease to the baseline Rtt105$^{Cy5}$ level. These data further support a model where Rtt105 does not completely dissociate from the RPA-$(dT)_{35}$ complex but is remodeled such that the N-terminal region of Rtt105 is moved away from DBD-D upon DNA binding. This movement likely encompasses movement of Rtt105 away from the trimerization core (DBD-C, RPA32, and RPA14; Tri-C) and towards the F-A-B region (OB-F, DBD-A and DBD-B) of RPA. Since ssDNA binding to RPA causes significant configurational changes, we next tested if free Rtt105 in solution can bind to preformed RPA-ssDNA complexes. FRET pairs were positioned on RPA (either RPA-DBD-A$^{Cy3}$ or RPA-DBD-D$^{Cy3}$) and Rtt105 (Rtt105$^{Cy5}$). Stoichiometrically bound RPA$^{Cy3}$-$(dT)_{35}$ complexes were

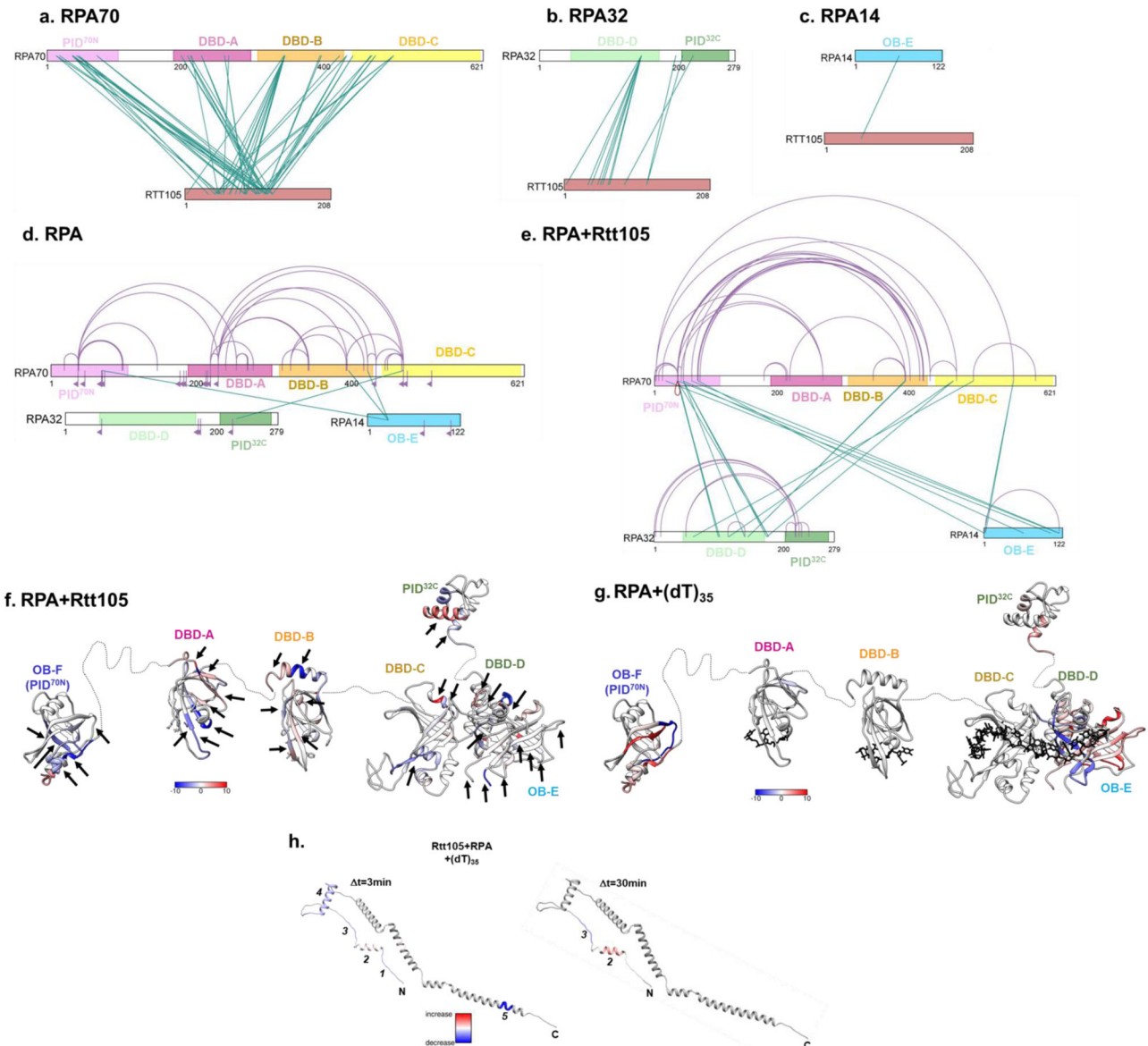

**Fig. 4 | Rtt105 configurationally staples RPA through multiple contacts with DBDs and PIDs.** Crosslinking mass spectrometry (XL-MS) analysis of the RPA-Rtt105 complex reveals crosslinks between Rtt105 and **a** RPA70, **b** RPA32, and **c** RPA14 subunits. Crosslinks are observed in all OB-folds and the BC-linker. Inter-subunit (green lines) and intra-subunit (purple traces) crosslinks in RPA are shown in the **d** absence or **e** presence of Rtt105, respectively. An increase in both inter- and intra-subunit crosslinks are observed upon Rtt105 binding to RPA. The inverted flag marks in **d** denote the sites of mono-crosslinks and show that they have no partner sites available in proximity for the BS3 crosslinker due to the extended

configuration of RPA in the absence of Rtt105. Hydrogen-deuterium mass spectrometry (HDX-MS) analysis of **f** RPA-Rtt105 and **g** RPA-(dT)$_{35}$ complexes show net changes in deuterium uptake or loss (ΔHDX) in almost all domains of RPA. The arrows point to ΔHDX that are unique to the Rtt105-RPA complex and not observed in the RPA-(dT)$_{35}$ complex. **h** ΔHDX changes in Rtt105 bound to RPA in the absence and presence of (dT)$_{35}$ are denoted in the AlphaFold derived model of Rtt105. The regions of change are denoted 1–5. The ΔHDX changes upon DNA binding suggests remodeling of Rtt105. The scales (blue to red) denote the net ΔHDX.

preformed and mixed with equimolar amounts of Rtt105$^{Cy5}$. In both cases, only a very small change in Cy5 fluorescence is observed (Supplementary Fig. 3a, b) suggesting that Rtt105 poorly binds to preformed RPA-DNA complexes. We do note that when RPA is in complex with ssDNA, its F-A-B is likely more exposed (described below) and thus, within the small fraction where Rtt105 binding is observed, we see a faster rate of Rtt105 binding to the F-A-B region (Supplementary Fig. 3c).

## Rtt105 configurationally staples RPA through contacts with multiple domains in RPA

A structure of the Rtt105-RPA complex is not available and our efforts to obtain one using cryo-electron microscopy have not been

successful because of the dynamic nature of the interactions. Thus, to understand how Rtt105 interacts with RPA and decipher how the complex is remodeled by ssDNA, we performed cross-linking mass spectrometry (XL-MS) and hydrogen-deuterium exchange mass spectrometry (HDX-MS) analysis of the Rtt105-RPA complex in the absence or presence of ssDNA. We observe good peptide coverage for both RPA and Rtt105 in MS analysis (92% RPA70, 67% RPA32, 74% RPA14, and 67% Rtt105; Supplementary Fig. 4) and thus can comprehensively assess the global conformational changes.

For XL-MS, RPA or RPA-Rtt105 were cross-linked with bis(sulphosuccinimidyl)suberate (BS3), which reacts with primary amines in lysine side chains and the N-termini[29]. RPA and Rtt105 cross-linked readily, as observed by the shift in the protein bands in SDS-PAGE

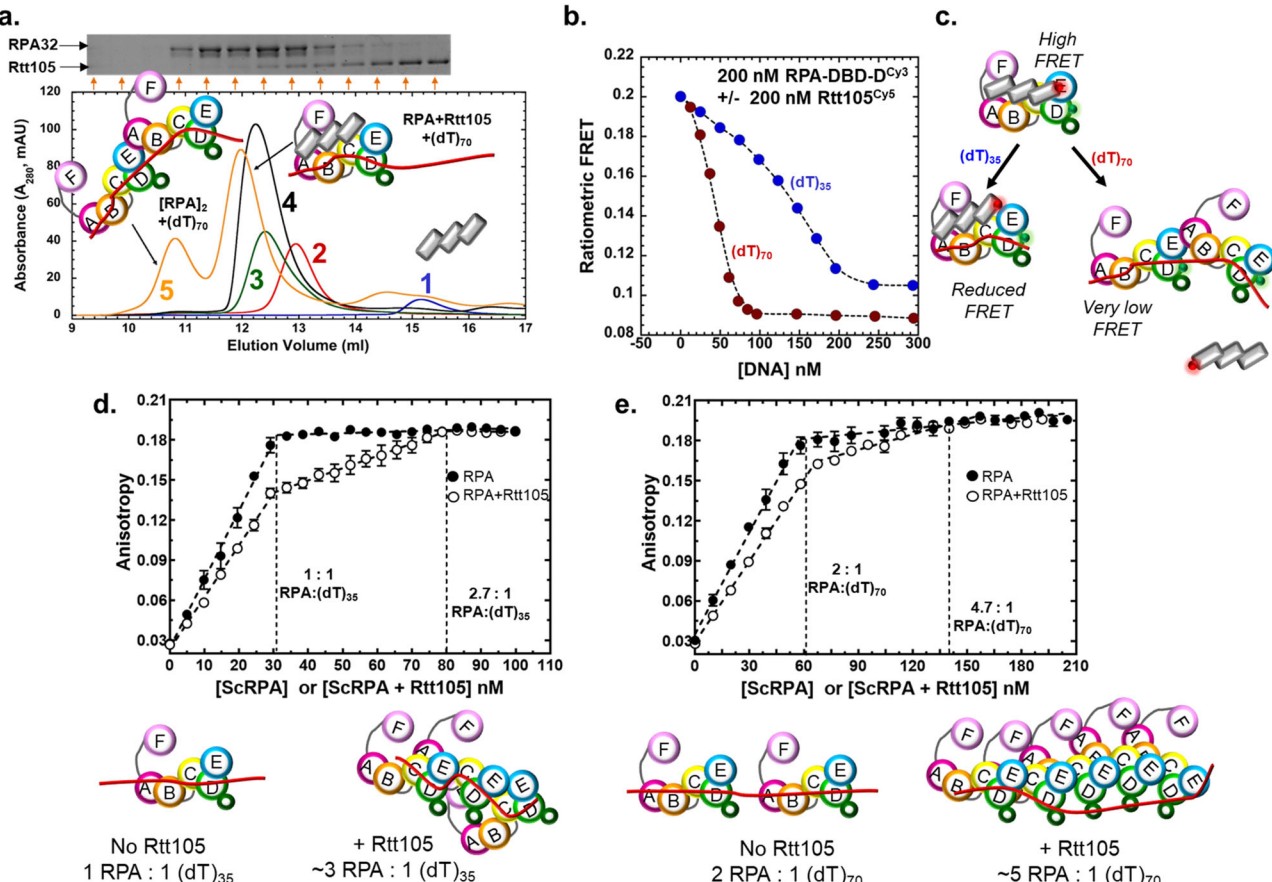

**Fig. 5 | Binding of multiple RPA molecules on longer ssDNA substrates triggers the release of Rtt105 and formation of a high-density RPA nucleoprotein filament. a** Size exclusion chromatography (SEC) analysis of the Rtt105-RPA complex in the presence of equimolar amounts of $(dT)_{70}$ show the presence of three distinct species. A larger $[RPA]_2$-$(dT)_{70}$ and smaller remodeled $[Rtt105$-$RPA$-$(dT)_{70}]^*$ complex are observed. Rtt105 is displaced from the larger complex and seen as the third peak. The distribution of Rtt105 in the three peaks can be noted in the SDS-PAGE gel shown as insert. Traces 1, 2, 3 and 4 are shown as reference and denote Rtt105, RPA, Rtt105-RPA, and the Rtt105-RPA-$(dT)_{35}$ complex, respectively. **b** Preformed RPA-DBD-D$^{Cy3}$-Rtt105$^{Cy5}$ complexes were mixed with increasing concentrations of either $(dT)_{35}$ or $(dT)_{70}$ and the change in FRET was measured. At ~2:1 ratios of RPA:$(dT)_{70}$

the FRET signal is significantly reduced. At ~1:1 ratio and lower, an alternate reduced FRET state is observed for $(dT)_{35}$. **c** The reduced FRET state likely corresponds to the remodeled $[Rtt105$-$RPA$-$(dT)_{35}]^*$ complex and the very low FRET state corresponds to the $[[RPA]_2$-$(dT)_{70}]$ state lacking Rtt105. **d** Fluorescence anisotropy experiments were performed with 30 nM $(dT)_{35}$ ssDNA by adding increasing amounts of RPA or the RPA:Rtt105 complex. On $(dT)_{35}$, RPA binds with 1:1 stoichiometry, but the Rtt105 increases the binding density of RPA to ~3 RPA/$(dT)_{35}$ molecule. **e** A similar phenomenon is observed on the longer $(dT)_{70}$ substrate where a higher density of RPA binding (~5 RPA/$(dT)_{70}$) is observed. Mean values +/− SE from $n = 3$ biologically independent experiments are denoted.

analysis after BS3 addition (Supplementary Fig. 5). MS analysis of the crosslinked peptides yielded several linkage pairs for the RPA and RPA-Rtt105 complexes, respectively (Fig. 4a–e). These data mapped onto the individual subunits of RPA reveal extensive crosslinks between the N-terminal two-thirds of Rtt105 and all DBDs and PIDs in RPA (Fig. 4a–c). Multiple contacts to OB-F, DBD-A, DBD-B, and DBD-C (all in RPA70) are observed. The F-A and A-B flexible linkers show no crosslinks, but a few are observed in the B-C linker (Fig. 4a). A limited number of crosslinks are observed in RPA32 with one peptide in DBD-D making 9 contacts in Rtt105 (Fig. 4b). Finally, a single crosslink is observed in OB-E (RPA14; Fig. 4c). These data agree with our observation that Rtt105 interacts more closely with the F-A-B half of RPA (Fig. 1e).

A closer look at the crosslinked peptides in the individual DBDs reveal that the crosslinks in DBD-A and DBD-B are not in the DNA binding pockets (Supplementary Fig. 6a, b). In contrast, some of the crosslinked peptides in DBD-C and DBD-D overlap with the DNA binding region (Supplementary Fig. 6c, d). Based on the predicted structure and charge distribution of Rtt105 (Supplementary Fig. 6e), the elongated helical regions likely extend and bind across the many domains of RPA. Circular dichroism (CD) analysis of Rtt105 confirm the

helical nature of Rtt105 (Supplementary Fig. 2b). Due to the extensive crosslinks observed we propose a configurational stapling model for the RPA-Rtt105 interaction. In this model, Rtt105 can be envisioned sitting on the surface of RPA, interacting with multiple domains, and configurationally constraining (or stapling) the DBDs and PIDs that are connected by flexible linkers.

Comparison of XL-MS data for RPA in the absence or presence of Rtt105 lends further support to the configurational stapling model. In the absence of Rtt105, a total of four inter-subunits crosslinks are captured between the three RPA subunits (Fig. 4d). In contract, Rtt105 binding induces 16 distinct inter-subunits crosslinks (Fig. 4e). These data show that the three subunits are brought in proximity upon Rtt105 binding. Analysis of the intra-subunit crosslinks further highlight the configurational stapling mechanism as the crosslinks between the individual DBDs and PIDs (within each subunit) are extensively increased in the presence of Rtt105 (Fig. 4d, e). Thus, in contrast to existing models that propose a stretching of the RPA domains upon Rtt105 binding[11,14,15], we show that the many DBDs and PIDs in RPA are compacted by Rtt105 through a configurational stapling mechanism.

To further assess the degree of configurational and conformational changes in RPA induced by Rtt105, we performed HDX-MS

analysis of RPA in the absence and presence of Rtt105 (Fig. 4f, g and Supplementary Figs. 7–11). In agreement with the XL-MS observations, changes in deuterium uptake/loss are observed in all the DBDs and PIDs of RPA. Most of the changes favor an increase in deuterium uptake (denoted as red) suggesting enhanced solvent exposure of RPA domains upon Rtt105 binding. To delineate whether Rtt105 binding overlaps with ssDNA binding to RPA, we compared the ΔHDX between the Rtt105-RPA and RPA-ssDNA complexes (Fig. 4f, g; regions marked with arrows are unique to Rtt105 binding). Thus, ssDNA ((dT)$_{35}$) binding remodels the Rtt105-RPA complex in agreement with the kinetic data.

Additional support for the remodeling of the RPA-Rtt105 complex by ssDNA is evident in the ΔHDX changes observed within Rtt105. Upon addition of DNA to the preformed Rtt105-RPA complex we observe deuterium uptake/release in five distinct regions, primarily around the N-terminal half (Fig. 4h and Supplementary Fig. 12). When assessed as a function of time, the remodeling of the N-terminal region of Rtt105 is observed at shorter time points. At later time points, the exchange patterns are preserved in only two of the original five regions. These data further support our model where ssDNA binding remodels the N-terminal region of Rtt105 away from Tri-C region of RPA and Rtt105 interactions with the RPA-ssDNA complex are maintained through interactions between Rtt105 and F-A-B region of RPA. The precise nature of the interactions will have to be established through future structural studies.

### Length of ssDNA and assembly of multiple RPA molecules promote release of Rtt105

In pull-down and single-molecule DNA curtain analysis Rtt105 has been proposed to not remain in complex with RPA on ssDNA[11,14,15]. Since we clearly observe a Rtt105-RPA-ssDNA complex on short ssDNA oligonucleotides (e.g. (dT)$_{35}$, Fig. 3a) we next tested whether ssDNA length influenced Rtt105 dissociation. Longer ssDNA substrates offer binding sites for more than one RPA molecule. In this scenario, we previously showed that DBD-A from one RPA can interact with DBD-E of the neighboring RPA while being stably bound to DNA through Tri-C-DNA interactions[30]. We hypothesized that since the F-A-B region is in an alternate configuration when multiple RPA are bound on DNA, Rtt105 might fully dissociate from the RPA-ssDNA complex. Using SEC, we tested whether Rtt105 remained bound to RPA when (dT)$_{70}$ ssDNA was added to the reaction (Fig. 5a). The elution profile shows three distinct species: [RPA]$_2$-(dT)$_{70}$, Rtt105-RPA-(dT)$_{70}$, and free Rtt105. The larger, earlier eluting, species does not contain Rtt105 (Fig. 5a and Supplementary Fig. 13). Thus, assembly of multiple RPA molecules serves as a trigger for Rtt105 dissociation.

To further probe the relationship between DNA length, multiple RPA binding, and Rtt105 dissociation, we used the FRET signal between RPA-DBD-D$^{Cy3}$ and Rtt105$^{Cy5}$ and followed complex dynamics as a function of either (dT)$_{35}$ or (dT)$_{70}$ concentration (Fig. 5b). Mixing 200 nM each of RPA-DBD-D$^{Cy3}$ and Rtt105$^{Cy5}$ yields a high FRET state, and the signal decreases as ssDNA is added to the reaction. (dT)$_{70}$ drives dissociation of Rtt105 at lower DNA concentrations (~100 nM) and produces a very low FRET state. At these ratios (2:1, RPA:(dT)$_{70}$) multiple RPA are expected to be bound to a single DNA molecule. In contrast, similar experiments done with (dT)$_{35}$ results in formation of an alternate low-FRET [Rtt105:RPA:(dT)$_{35}$]* state that corresponds to the remodeled complex where most Rtt105 remains bound, but in a remodeled state. Furthermore, the longer (dT)$_{70}$ substrate is 3-fold better at arriving at the low FRET state compared to (dT)$_{35}$ (K$_{1/2}$ = 130.4 ± 10.5 and 42.8 ± 0.37 nM for (dT)$_{35}$ and (dT)$_{70}$, respectively; Fig. 5b). Performing this experiment as a function of ssDNA length reveals that ssDNA ~35 nt and longer are effective in promoting Rtt105 dissociation from RPA (Supplementary Fig. 14). Thus, when adequate ssDNA is present to saturate all the individual DBDs (≤ (dT)$_{35}$) within one RPA molecule, Rtt105 is

remodeled. Longer DNA (≥(dT)$_{35}$), and higher RPA:DNA ratios promote binding of multiple RPA molecules and drive Rtt105 dissociation (Fig. 5c).

### Rtt105 drives formation of high-density RPA-ssDNA filaments

In single molecule DNA curtain experiments, Rtt105 was shown to drive formation of stretched RPA-ssDNA filaments[11]. Since Rtt105 was proposed to stretch out the domains of RPA, the model invoked elongated RPA bound to longer ssDNA with Rtt105 not bound[15]. Our data show a compaction of RPA by Rtt105. To reconcile these differences, we wondered if Rtt105 promoted alternate DNA-bound RPA configurations. For example, if Rtt105 were bound to the F-A-B part of RPA, one could envision an RPA-ssDNA filament where multiple RPA are bound using only the Tri-C region. Since the Tri-C region is more stably bound to ssDNA[20], the resulting filament could be more rigid and elongated as observed in the DNA curtain experiments. To test this model, we used fluorescence anisotropy to monitor RPA binding to (dT)$_{35}$ or (dT)$_{70}$ ssDNA in the absence or presence of Rtt105. On the shorter (dT)$_{35}$ substrate, the signal saturates at 1:1 (RPA:DNA) in the absence of Rtt105 (Fig. 5d). In contrast, the ratio shifts to 2.7:1 (RPA:DNA) when the RPA-Rtt105 complex is used (Fig. 5d). On the longer (dT)$_{70}$ substrate, this effect is further exaggerated as ~5 molecules of RPA are loaded onto the ssDNA in the presence of Rtt105 compared to ~2 when no Rtt105 is present in the reaction (Fig. 5e). Thus, remarkably, the configuration of RPA on ssDNA formed in the presence of Rtt105 is stable and distinctly different with a higher density of RPA molecules bound to the ssDNA. Interestingly, the difference in the amplitude of the anisotropy signals at the 1:1 saturation point for both ssDNA substrates suggest that the hydrodynamic radius of the RPA-DNA complex in the absence of Rtt105 is larger. We propose that the RPA molecules formed in the presence of Rtt105 likely have the F-A-B region (the dynamic half of RPA) not bound to DNA.

In all the above experiments, we investigated modulation of the RPA-Rtt105 complex using short ssDNA oligonucleotides. To test whether our observations of RPA-Rtt105 remodeling were recapitulated on longer kilobase-long ssDNA we utilized C-trap analysis. Here lambda DNA (~48.5 kbp) is tethered between two beads and stretched to produce ssDNA (Supplementary Fig. 15a–c). Fluorescent proteins binding to DNA, remodeling, and/or dissociating are visualized as kymographs as a function of time. Rtt105$^{Cy5}$ alone does not interact with DNA at either low or high concentrations and thus no fluorescent spots are observed (Supplementary Fig. 15d). When the Rtt105$^{Cy5}$-RPA complex is premixed at 1:1 ratio and first incubated with the DNA in the first chamber and then moved to chamber containing buffer, a few spots of Rtt105$^{Cy5}$ are occasionally encountered (Supplementary Fig. 15e, f). Even in these rare instances, Rtt105$^{Cy5}$ dissociates from these spots with a t$_{1/2}$~5 s. Quantitation of these data over multiple experiments reveal a distribution of 1–4 binding/dissociation events per kymograph recorded (Supplementary Fig. 15f). These experiments provide direct visual evidence for engagement of Rtt105-RPA onto DNA and Rtt105 dissociation under conditions where multiple RPA can bind onto a single DNA lattice and form a nucleoprotein filament.

### Rtt105 impedes facilitated exchange of RPA

Free RPA has been shown to exchange with RPA bound to ssDNA through a process called facilitated exchange (FE)[31]. This activity arises from the dynamic binding, rearrangement, and dissociation of one or more DBDs on the DNA and thus transiently exposed short segments of ssDNA allow free RPA to gain access. In this scenario, while multiple RPA molecules are bound on ssDNA, we expect only short segments DNA exposed at any given point during FE. The biological significance of FE is poorly understood but thought to contribute to RPA dynamics and RPA-promoted enhancement or impediment of DNA metabolic processes in the cell. FE could also be

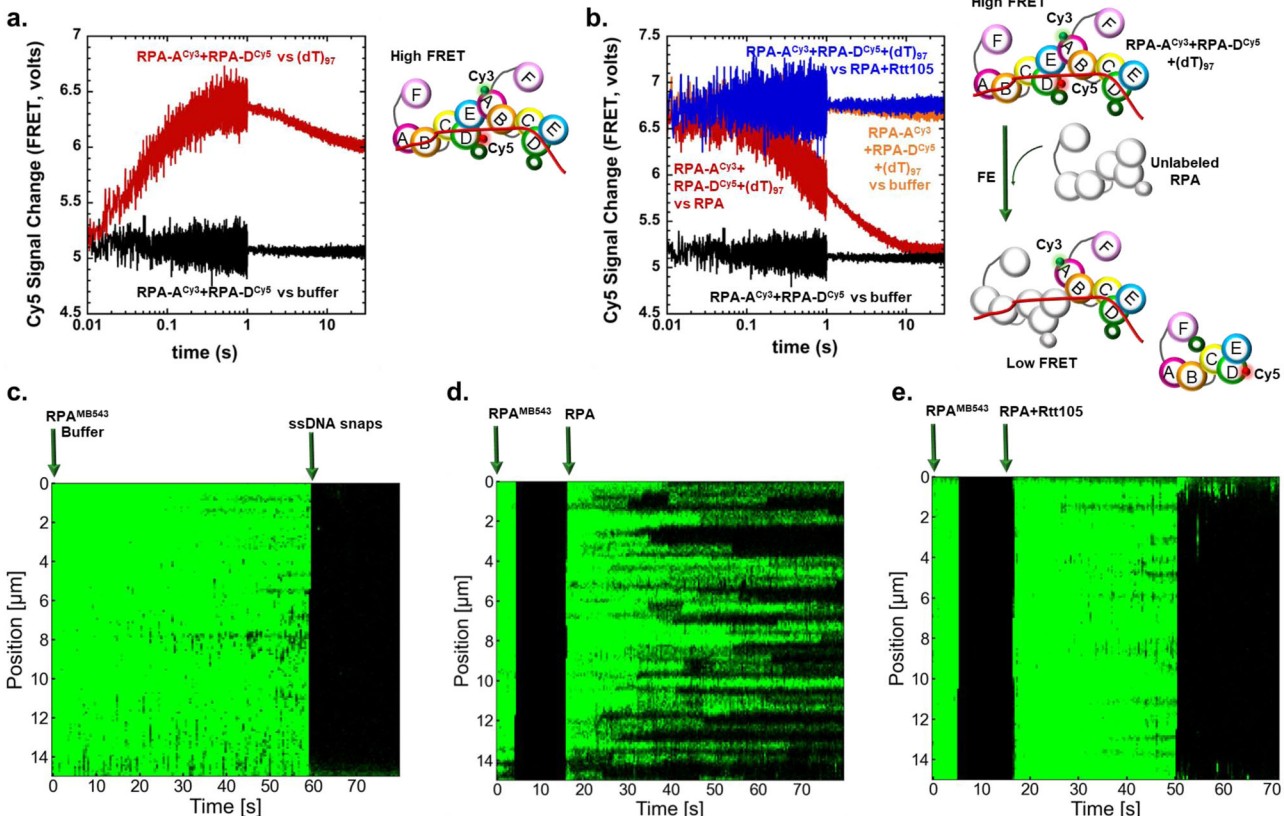

**Fig. 6 | Rtt105 inhibits facilitated exchange activity of RPA. a** A high FRET complex is formed when multiple molecules of RPA-DBD-A$^{Cy5}$ and RPA-DBD-D$^{Cy3}$ bind to a (dT)$_{97}$ substrate (red). FRET-induced Cy5 fluorescence is not observed in the absence of DNA (black). **b** Facilitated exchange (FE) activity was measured by mixing preformed [RPA-DBD-A$^{Cy5}$:RPA-DBD-D$^{Cy3}$:(dT)$_{97}$] complexes with unlabeled RPA (red) resulting in a loss of Cy5 fluorescence. Rtt105 inhibits the FE activity as preformed Rtt105:RPA complexes do not perturb the Cy5 signal (blue). Control experiments in the absence of unlabeled RPA (orange) or DNA (black) are also shown as reference for the high and low FRET-induced Cy5 fluorescence states. Models for FE are denoted on the right. Representative stopped flow data averaged from seven to eight shots from one experiment are shown. **c** C-trap experiments to investigate FE of RPA were performed on ssDNA that was generated by mechanically unfolding lambda dsDNA (~48.5kbp) in the optical trap and subsequently incubated with RPA-DBD-D$^{MB543}$ (10 nM) for 15–20 s and stably formed fluorescent RPA coated ssDNA nucleoprotein filaments are observed. **d** The fluorescent RPA nucleoprotein filament was subsequently moved to a channel containing unlabeled RPA (10 nM) and after a 10 s incubation in the dark FE can be observed. Spots where loss of fluorescence is observed are events where the unlabeled RPA have replaced fluorescent RPA molecules. **e** When challenged with unlabeled RPA:Rtt105, no loss in fluorescence is observed. Thus, Rtt105 inhibits the FE activity of RPA. C-trap data traces from one-DNA molecule per condition are shown, but more than 10 DNA molecules were visualized and recorded for analysis.

envisioned as a counterproductive process in the cell where excess RPA perturbs the stability of an RPA nucleoprotein filament. We hypothesized that Rtt105 might influence FE by serving as a sink to sequester free RPA. To test this idea, we performed FE experiments using RPA labeled with FRET pairs. Binding of equimolar amounts of RPA-DBD-A$^{Cy3}$ and RPA-DBD-D$^{Cy5}$ (100 nM each) to (dT)$_{97}$ ssDNA (30 nM) results in a high FRET signal as multiple RPA molecules assemble (Fig. 6a). When excess unlabeled RPA is added to a pre-formed RPA-DBD-A$^{Cy3}$:RPA-DBD-D$^{Cy5}$:(dT)$_{97}$ complex, FE occurs leading to a loss in the FRET signal (Fig. 6b, red trace). In contrast, when challenged with RPA prebound to Rtt105, no FE is observed, and the FRET signal is not perturbed (Fig. 6b, blue trace). Thus, the short segments of free ssDNA available during FE does not allow binding of the Rtt105·RPA complex. To further investigate whether Rtt105 blocks FE on long RPA nucleoprotein filaments, we used C-trap analysis where lambda DNA (~48.5 kbp) was first coated with fluorescent RPA (RPA-DBD-D$^{MB543}$) and then moved over to a channel containing either buffer only, buffer with unlabeled RPA, or unlabeled RPA plus Rtt105 (Fig. 6c–e). Extensive FE is observed when unlabeled RPA is introduced as evidence by the loss in fluorescence. When challenged with the RPA + Rtt105 complex, no FE is observed. Thus, Rtt105 blocks FE activity of RPA and thus likely contributes to the formation of stable RPA nucleoprotein filaments.

## Complex formation with Rtt105 negatively regulates interactions with RPA-interacting proteins like Rad52

Since Rtt105 configurationally staples RPA and restricts the DBDs and PIDs, we wondered if the protein-protein interaction activities of RPA were regulated by Rtt105. This could prevent spurious RPA interactions with its myriad binding partners upon nuclear import from the cytoplasm and add to the functions of Rtt105 in genomic integrity. To test this hypothesis, we tested whether Rtt105 influenced RPA binding to Rad52, a mediator protein that physically interacts with RPA and functions to promote homologous recombination (HR)[32]. SEC has limitations in resolving larger molecular weight complexes, thus we used analytical ultracentrifugation (AUC) to monitor complex formation. Rtt105, RPA, Rad52 all sediment as distinct complexes (Fig. 7a). RPA and Rad52 form a complex in the absence or presence of ssDNA (dT)$_{35}$ (Fig. 7a). We and others have previously shown that Rad52 interacts with RPA on and off the DNA and these data agree with our knowledge of these interactions[17,32,33]. The only key difference is the oligomeric state of yeast Rad52 which is widely considered to be a heptamer[34]. In our ongoing cryoEM studies, *S. cerevisiae* Rad52 is a homodecamer (Deveryshetty and Antony, unpublished data) and the stoichiometries denoted here reflect this finding. In the absence of ssDNA, Rtt105 remains as a complex with RPA and prevents interaction with Rad52 (Fig. 7b). Thus, Rtt105 functions as a negative regulator of RPA-Rad52 interactions in the

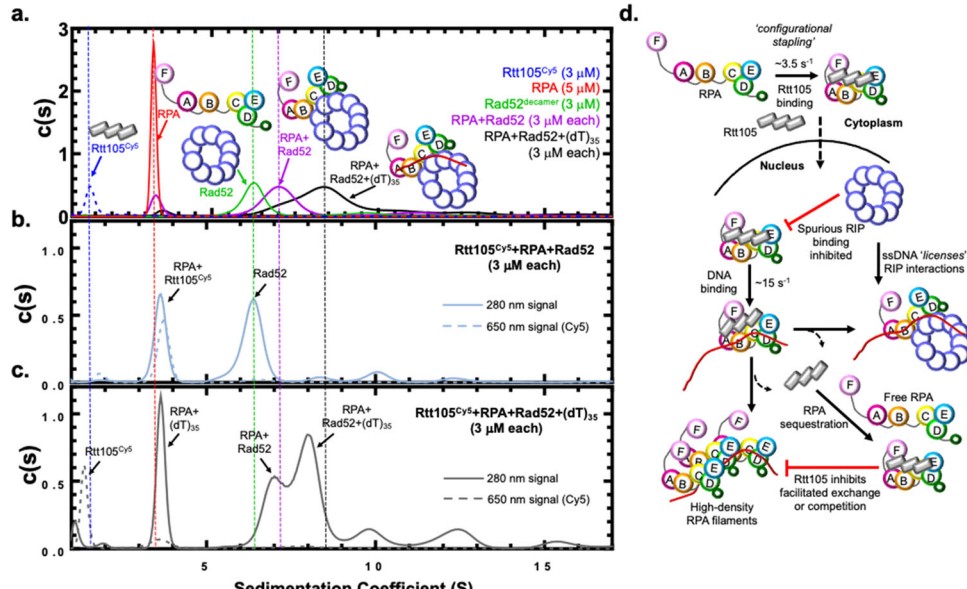

**Fig. 7 | Rtt105 inhibits spurious RPA-Rad52 interactions in the absence of ssDNA. a** Analytical ultracentrifugation (AUC) analysis of RPA, Rtt105$^{Cy5}$, Rad52, and the RPA-Rad52 and RPA-Rad52-(dT)$_{35}$ complexes show them sedimenting as distinct complexes. The 280 nm signals were monitored for all the proteins and complexes except Rtt105$^{Cy5}$ which was monitored at 650 nm (dashed line). The horizontal dotted lines serve as a visual guide for the sedimentation positions of the denoted proteins across all the AUC experiment panels. **b** Rtt105$^{Cy5}$, RPA, and Rad52 were mixed in equimolar concentrations and analyzed. In the presence of Rtt105, Rad52 does not interact with RPA and sediments independently while RPA and Rtt105 remain as a complex. **c** When an equimolar amount of ssDNA [(dT)$_{35}$] is present, Rtt105 is displaced and sediments similar to free Rtt105. RPA and Rad52 partition into multiple RPA+DNA, RPA+Rad52, and RPA+Rad52+DNA complexes.

Importantly, Rtt105 is not part of most of these complexes. A small fraction of the Rtt105 is present along with the RPA-DNA complex. Representative data from a single run are shown, but repeated $n = 3$ with biological replicates. **d** Rtt105 configurationally staples RPA through interactions with multiple domains including the DNA binding and protein interaction domains. The complex is shuttled into the nucleus where promiscuous interactions with RPA binding proteins such as Rad52 are inhibited by Rtt105 in the absence of ssDNA. Availability of ssDNA triggers configurational and conformational changes within the RPA-Rtt105 complex and this higher order complex can interact with RIPs such as Rad52 or drive assembly of multiple RPA-bound DNA structures. Both these scenarios drive dissociation of Rtt105. Rtt105 can also sequester free RPA to prevent untimely facilitated exchange (FE) events as the RPA-Rtt105 complex cannot perform FE.

absence of DNA. In the presence of ssDNA Rtt105 is released and formation of RPA-DNA, RPA-Rad52, and RPA-Rad52-DNA complexes are observed (Fig. 7c). These data suggest that ssDNA acts as a licensing agent to release Rtt105 from RPA and promote RPA interactions with Rad52, and likely with other RPA-interacting proteins (Fig. 7d).

## Discussion

Rtt105 functions akin to a chaperone-like protein to transport RPA into the nucleus, and deletion of Rtt105 reduces RPA availability for DNA metabolic roles including DNA repair and recombination[11,13–15]. Previous studies have proposed a model where Rtt105 stretches the many domains of RPA to enhance DNA binding[15]. Here, we present a modified model for Rtt105 mediated regulation of RPA. First, our findings show extensive contacts between Rtt105 and RPA that promotes a compaction of the four DNA binding and two protein-interaction domains of RPA. The CL-MS and HDX MS analysis show configurational and conformational changes in all domains and a few of the intervening flexible linkers. While a structure of Rtt105 or the RPA-Rtt105 complex is not yet available, the AlphaFold prediction renders a helical protein that could be stretched over a large radius (Fig. 1b). Thus, we propose that configurational stapling compacts RPA through extensive contacts with Rtt105 and promotes nuclear import (Fig. 7d). Rtt105 is an extensively charged protein and stretches of negatively charged patches are observed that likely bind close to the positively charged DNA binding pockets in RPA. In support of this model, we see differences in the DNA binding properties of the Rtt105-RPA complex.

While Rtt105 does not bind directly to DNA, Rtt105 complexed with RPA influences the kinetics and conformations/configurations of the DBDs on ssDNA. Positioning of site-specific fluorophores on either DBD-A or DBD-D enabled us to quantify how each of these domains

bind to DNA and are remodeled. Rtt105 reduces the rate of ssDNA binding by DBD-D by ~2-fold but does not change the final configuration of this domain on DNA. In contrast, Rtt105 does not change the rate of ssDNA binding to DBD-A, but significantly alters the final configuration. Thus, we envision Rtt105 situated on both the Tri-C and FAB halves of RPA in the absence of ssDNA as supported by the XL-MS analysis (Fig. 4). The region close to Tri-C occludes the DNA binding site in DBD-D (possibly in DBD-C as well) and is remodeled in the presence of ssDNA. The Rtt105 region close to the FAB region likely does not block DNA binding but restrains the configurational freedom of those domains. In the [Rtt105-RPA-(dT)$_{35}$] complex, ssDNA must be bound close to the Tri-C region. Further evidence for this model arises from the dissociation of Rtt105 on ssDNA longer than (dT)$_{35}$ where multiple RPA can bind. Here, DBD-A (RPA70) from one RPA contacts the OB-E (RPA14) of the neighboring RPA[30]. This interaction likely triggers the dissociation of Rtt105. Moreover, Rtt105 also promotes formation of RPA nucleoprotein filaments that have a higher density of RPA molecules bound. This property renders the RPA nucleoprotein filament more stable and explains the additional stretching of ssDNA observed in DNA curtain experiments[11]. In the cell, formation of such RPA nucleoprotein platforms triggers the DNA damage response[3] and Rtt105 might contribute to such functions by not releasing RPA until longer stretches of ssDNA become exposed.

Contradictory to earlier studies[11,14], we do not see observable differences in the $K_D$ for RPA ssDNA interactions in the presence of Rtt105. This is not surprising as RPA will be macroscopically bound with high affinity to ssDNA while select DBDs can undergo dynamic rearrangements[18]. Moreover, because of the high ssDNA binding affinity ($K_d < 10^{-10}$M), enhancements of DNA binding affinity cannot be measured using traditional bulk ensemble measurements. Our in vivo

studies show a distinction between functions associated with Rtt105 and DNA interactions of RPA. Loss of Rtt105 (*Δrtt105*) or reduction in RPA-ssDNA interactions (*rfa1 zm1, zm2,* or *t33* mutations) shows defects in cell growth and a combination of *Δrtt105* and *rfa1* mutations result in severe growth defects. The additive nature of these phenotypes again suggests that Rtt105 does not directly influence the macroscopic ssDNA binding properties of RPA, but regulates it through either affecting nuclear cytoplasmic shuttling, RIP interactions, or processes such as facilitated exchange and the differences in the density of RPA bound to ssDNA. Rtt105 contributes to the levels of RPA in the nucleus, however, no severe defects in global DNA synthesis are observed under non-stressed conditions[13]. In contrast, at perturbed replication forks, the lack of Rtt105 becomes important for RPA loading[13]. In this scenario, longer ssDNA intermediates are generated and thus, the inhibition of FE by Rtt105 and formation of higher density RPA binding could explain the formation of more stable RPA filaments.

Since RPA interactions with Rtt105 occur in the cell well before the availability of ssDNA, we hypothesize that posttranslational modifications of RPA (and/or Rtt105), such as phosphorylation by kinases, might influence the interaction. In support of this idea, a phospho-mimetic of RPA carrying a Ser to Asp substitution at position 178 in RPA70 (RPA$^{S178D}$) shows reduced binding to Rtt105 (Supplementary Fig. 16). Rtt105 has also been implicated in the resolution of G-quadruplex structures by RPA[35]. In vitro, we do not see any direct Rtt105 enhancement of RPA promoted G-quadruplex unwinding (Supplementary Fig. 17). Thus, in this scenario, we propose that Rtt105 regulates the activity of RPA by controlling its nuclear transport, inhibiting RIP interactions in the absence of ssDNA, and regulating the availability of free RPA.

Rtt105 appears to block spurious interactions with RIPs in the absence of ssDNA (Fig. 7a). Such protein-protein interactions are primarily mediated by either OB-F in RPA70 (PID$^{70N}$) or the winged helix domain in RPA32 (PID$^{32C}$)[18]. A few interacting proteins such as Rad52 are composite binders that interact with a second region in RPA[8,32,36,37]. In the case of Rad52, interactions between PID$^{32C}$ and a buried site in DBDs-A & B have been identified[8,32,36-38]. Rad52 is a mediator protein that facilitates the nucleation of the Rad51 recombinase on RPA-coated ssDNA[39]. Rad52 selectively remodels DBD-D in RPA32 and gains access towards the 3′-end of the resected ssDNA during homologous recombination[17,18]. Upon Rtt105 binding, PID$^{70N}$ is positioned closer to DBD-B and DBD-C (Fig. 4e) and thus is likely occluded from interacting proteins that use this region for binding. Similarly, new crosslinks are observed between PID$^{32C}$ and the N-terminal region of RPA32 in the presence of Rtt105 (Fig. 4e). Thus, we propose that the configurational stapling of RPA by Rtt105 extends to the protein-protein interaction domains. Functionally, Rtt105 binding protects RPA from binding to potential RIPs upon nuclear localization. When ssDNA is exposed, removal/remodeling of Rtt105 occurs which in turn likely releases the protein-interaction domains and leads to recruitment of RIPs onto the RPA-coated ssDNA. The extent of release could also be facilitated by posttranslational modifications of RPA such as phosphorylation (Supplementary Fig. 16).

Finally, in higher eukaryotes, RPAIN (RPA interacting protein) or RIPα (RPA interacting protein α) is proposed to be the functional ortholog of Rtt105 as they share poor sequence similarity[14]. The AlphaFold predicted structure of RPAIN shows a reasonable degree of resemblance to Rtt105 (Supplementary Fig. 18). RPAIN has also been proposed to function by enhancing the DNA binding activity of human RPA (hRPA)[14]. Similar to our observations for Rtt105 and yeast RPA, we see complex formation between hRPA and RPAIN, but do not observe a stimulation of the macroscopic ssDNA binding properties (Supplementary Fig. 18). Interestingly, shorter DNA (dT)$_{35}$ can promote dissociation of the RPAIN-hRPA complex, and future work will focus on deciphering the differences in the mechanisms of action of RPAIN.

## Methods

### Reagents and buffers
Chemicals were purchased from Sigma-Millipore Inc., Research Products International Inc. and Gold Biotechnology Inc. Fluorescent and unlabeled oligonucleotides were synthesized by Integrated DNA Technologies. Enzymes for molecular biology were purchased from New England Biolabs. Resins for protein purification were sourced from GE-Cytiva Life Sciences Inc. Fluorophores for protein labeling were from Click Chemistry Tools Inc.

### Plasmids for protein overproduction
An RSF-Duet1 plasmid coding for Rtt105 with a N-terminal 6x-poly-histidine tag was used. Mutations in *Rtt105* were introduced using site-directed mutagenesis. Plasmids for RPA and 4-azidophenylalanine (4AZP) incorporation were as described[17,22,23]. A codon-optimized open reading frame for human RPAIN was synthesized (Genescript Inc.) and carries a SUMO protease cleavable N-terminal Strep-6xHIS-SUMO tag.

### Purification of RPA, Rtt105, and RPAIN
Rtt105 was overproduced in Rosstta-2 PlysS *E. coli* cells and purified as described[11]. *Saccharomyces cerevisiae* and human RPA were purified as described[22,40]. Non-canonical amino acid (4AZP) incorporation based fluorescently-labeled RPA were generated as described[17,22,23]. Concentration of Rtt105 was determined spectroscopically using $\varepsilon_{280} = 18,450\ M^{-1}cm^{-1}$. Rtt105 was flash frozen and stored at −70 °C. Concentration of unlabeled and labeled RPA was measured spectroscopically using $\varepsilon_{280} = 98,500\ M^{-1}cm^{-1}$. Labeling efficiency for fluorescent RPA was calculated as described using absorption values measured at 280 nm and $\varepsilon_{280} = 98500\ M^{-1}cm^{-1}$ for RPA, at 550 nm with $\varepsilon_{550} = 105,000\ M^{-1}cm^{-1}$ for RPA-MB543, at 555 nm with $\varepsilon_{555} = 150,000\ M^{-1}cm^{-1}$ for RPA-Cy3, and at 650 nm with $\varepsilon_{650} = 250,000\ M^{-1}cm^{-1}$ for RPA-Cy5 versions[17,23]. Human RPAIN was overproduced in Rosstta-2 PlysS *E. coli* cells by inducing the cells at $OD_{600} = 0.6$ with 1 mM IPTG and growing them overnight at 18 °C. The cell pellets were lysed in buffer containing 30 mM HEPES pH 7.8, 300 mM KCl, 1 mM TCEP-HCl, 10 % v/v glycerol, 0.04 mg/ml of lysozyme and protease inhibitor cocktail and then sonicated on ice for a total of 2 min. After centrifugation at 4 °C for 60 mins at 41,107 g, the clarified lysate was batch bound to Ni$^{2+}$-NTA beads for 18 h at 4 °C. Beads were washed sequentially with wash buffer (30 mM HEPES pH 7.8, 0.02% Tween-20, 10 mM imidazole, 1 mM TCEP-HCl, 10% v/v glycerol, and protease inhibitor cocktail) containing varying amounts of KCl (0.3 M, 2 M and 0.05 M). RPAIN was eluted with elution buffer (30 mM HEPES pH 7.8, 50 mM KCl, 0.02% Tween-20, 400 mM imidazole, 1 mM TCEP-HCl, 10% v/v glycerol, and protease inhibitor cocktail). RPAIN containing fractions were pooled and further fractionated using a 10 mL Q-sepharose Fast Flow column (Cytiva Life Sciences) equilibrated in buffer (30 mM HEPES pH 7.8, 0.02% Tween-20, 0.1 M KCl, 0.25 mM EDTA pH 8.0, 1 mM TCEP-HCl, 10% glycerol, and protease inhibitor cocktail). After loading and washing, RPAIN was eluted in the same buffer with a 0.1–1.5 M KCl gradient. RPAIN containing fractions were pooled and digested for 18 h at 4 °C with SUMO protease (1:10 ratio) to remove the N-terminal tag. Cleaved RPAIN was separated using a Hi-Load 16/600 Superdex 200 pg size exclusion column (Cytiva Life Sciences) using buffer (30 mM HEPES pH 7.8, 300 mM KCl, 0.02 % Tween-20, 0.25 mM EDTA pH 8.0, 1 mM TCEP-HCl, and 10 % v/v glycerol). RPAIN containing fractions were pooled, concentrated using a spin-concentrator, flash frozen, and stored at −80 °C. RPAIN concentration was determined spectroscopically using extinction coefficient $\varepsilon_{280} = 28,585\ M^{-1}\ cm^{-1}$.

### Generation of fluorescently-labeled Rtt105
Rtt105 has two consecutive Cys residues at positions 12 and 13, respectively. Either or both Cys residues can be substituted with Ser without loss of binding to RPA (Supplementary Fig. 2). We used Cys-

**Table 1 | *S. cerevisiae* strains used in this study**

| Strain | Genotype | Source |
|---|---|---|
| X8584-2A | *Rtt105-TAP::TRP* | This study |
| X8049-7B | *MATα rfa1-zm1* | Ref. 24 |
| X8047-1B | *MATα rfa1-zm2* | Ref. 24 |
| X8584-2C | *Rtt105-TAP::TRP rfa1-zm1* | This study |
| X8585-11C | *Rtt105-TAP::TRP rfa1-zm2* | This study |
| G996 | *MATα rfa1-t33* | L. Symington |
| X8587-1A | *Rtt105-TAP::TRP rfa1-t33* | This study |
| T2159-16 | *MATa rtt105Δ::hphMX6* | This study |

12 for attachment of fluorophores using maleimide chemistry and converted Cys-13 to Ser. This version of Rtt105 (Rtt105$^{C13S}$) was purified similar to the wild-type protein and ~5 ml of 100 µM Rtt105$^{C13S}$ was dialyzed extensively in labeling buffer (30 mM HEPES, pH 7.8, 200 mM KCl, 0.25 mM EDTA, pH 8.0, 0.01 % Tween-20, and 10 % v/v glycerol) to remove from the storage buffer. After 4 buffer exchanges, a 1.5-fold molar excess of Cy5-maleimide dye was added to the dialyzed protein and incubated at 4 °C for 3 h. The reaction was then quenched with 0.5 % β-mecraptoethanol (βME). Excess dye was separated from the protein using a Biogel-P4 column (Bio-rad laboratories) and resolved with labeling buffer with 2 mM TCEP. The concentration of Rtt105 calculated spectroscopically using extinction coefficient $ε_{280} = 18,450$ $M^{-1}cm^{-1}$ and labeling efficiency was calculated using the Cy5 absorbance signal and $ε_{650} = 250,000$ $M^{-1}cm^{-1}$. Rtt105 absorbance values at 280 nm were also corrected for minor signal interference from Cy5 by measuring the percent contribution of free Cy5.

### Yeast strains and genetic techniques
Standard procedures were used for cell growth and media preparation. Strains used are provided in Table 1 and are isogenic to W1588-4C, a *RAD5* derivative of W303 *(MATa ade2-1 can1-100 ura3-1 his3-11,15, leu2-3, 112 trp1-1 rad5-535)*[41]. Gene deletion to generate *rtt105Δ* strain was performed following standard PCR based method. Standard yeast genetic procedures were used for tetrad analyses and at least two biological duplicates were used for each genotype.

### Co-Immunoprecipitation
Yeast cells from log phase cultures growing in YPD were harvested and lysed by bead beating in TMG-140 buffer (10 mM Tris-HCl, pH 8.0, 4 mM MgCl$_2$, 10% v/v glycerol, 140 mM NaCl, 0.1 mM EDTA, 0.5% Tween20, 1 mM DTT and Roche cOmplete-Ultra EDTA free protease inhibitor). DNA was digested by incubation with benzonase for 30 min at 4 °C. Lysates were cleared by centrifugation and incubated in TMG-140 buffer with IgG sepharose beads for 2 h at 4 °C. After incubation, beads were washed with TMG-140 and proteins were eluted with Laemmli buffer. Proteins were separated on gradient gels followed by western blotting with antibodies against Rfa1 (a kind gift from Dr. Steven Brill at Rutgers University). Rfa1 primary antibody was used at 1:6000 dilution. Anti-rabbit HRP secondary body was used at 1:8000 dilution (VWR Scientific).

### Secondary structure analysis using circular dichroism (CD)
CD measurements were used to compare the secondary structures of Rtt105 and the Cys variants of Rtt105. A nitrogen-fused observation chamber with a cell pathlength of 10 mm was used. All CD traces were obtained between 200–260 nm at 20 °C on a Chirascan CD spectrometer (Applied Photophysics Inc. using Pro-data Chirascan software). 600 nM of Rtt105$^{WT}$, Rtt105$^{C13S}$, Rtt105$^{C12S}$, Rtt105$^{CCSS}$ in CD reaction buffer (5 mM Tris-Cl pH 7.8, 100 mM KCl, 5 mM MgCl$_2$, and 6% v/v glycerol) was used to obtain CD spectra. The results were collected using 1 nm step size, 1 nm bandwidth and 5 traces were averaged.

### Analysis of complex formation using size exclusion chromatography (SEC)
600 µl of the noted concentrations of RPA, Rtt105, or the complex in the absence or presence of equimolar amounts of DNA were resolved on a 10/300 Superose 6 Increase column using an AKTA-pure FPLC system. Protein and protein-DNA complexes were incubated at 4 °C for 10 min before analysis. Resolution was performed using Rtt105-SEC buffer (30 mM HEPES, pH 7.8, 100 mM KCl, 1 mM TCEP-HCl, and 10% v/v glycerol). A total of 30 ml elution volume was collected as 0.5 ml fractions and further analyzed on 10% SDS-PAGE.

### Measurement of RPA-ssDNA interactions using electrophoretic mobility band shift analysis (EMSA)
10 nM 5'-Cy5-(dT)$_{30}$ ssDNA was incubated with indicated amounts of RPA at 25 °C in EMSA buffer (25 mM Tris-Cl, pH 7.5, 200 mM NaCl, 5 mM MgCl$_2$, 5% v/v glycerol, and 0.05% Tween-20) for 10 min. For experiments performed in the presence of Rtt105, RPA and Rtt105 were premixed in equimolar ratios at 25 °C for 5 min before DNA was introduced. The reaction mixture (20 µl) was mixed with 10 µl of 70% v/v glycerol, mixed, and loaded onto a 6–15% bis-acrylamide gradient gel and resolved using 1x TBE buffer. The gels were scanned using an iBright 1500 imager (Thermo Fischer Inc.) and the Cy5 fluorescence associated with ssDNA (unbound fraction) or ssDNA+protein (bound fraction) bands was background subtracted and quantified with the associated iBright software. The mean values and standard deviation from three independent experiments were plotted for analysis. K$_D$ was estimated by nonlinear least squares fitting to Eq. (1):

$$F_b = \frac{F_{max}[RPA]^h}{K_D{}^h + [RPA]^h} \qquad (1)$$

$F_b$ is the fraction of bound ssDNA determined from fluorescence intensities of two bands, namely:

$$F_b = \frac{bound}{bound + unbound}$$

$F_{max}$ is the fraction of bound ssDNA at saturating protein concentration, $K_D$ is the apparent dissociation constant, [$RPA$] is concentration of RPA or RPA+Rtt105 (1:1) in each well, and $h$ is the Hill coefficient.

### Measurement of RPA and ssDNA binding in presence or absence of Rtt105 using fluorescence anisotropy
5'-FAM-(dT)$_{35}$ ssDNA was diluted to 10 nM in 1x RPA reaction buffer (30 mM HEPES pH 7.8, 100 mM KCl, 6 % v/v glycerol, 5 mM MgCl$_2$, and 1 mM βME). 180 µl of this working stock was added to a 3 mm pathlength quartz cuvette (Starna Cells Inc.) and the temperature was maintained at 23 °C. Fluorescence anisotropy of the FAM-labeled ssDNA was measured using PC1 spectrofluorometer (ISS Inc.) and data collected using the associated Vinci 3 software. Samples were excited at 488 nm and the resulting emission was collected using a 520 nm band pass emission filter. Five consecutive anisotropy readings were acquired from ssDNA alone or after stepwise addition of RPA alone, or 1:1 stock of RPA and Rtt105 (each protein at 5 µM). The concentrations of ssDNA, the added protein, and the reduction in intensity were corrected for effects due to dilution alone. Measured anisotropy values were corrected for the G-factor, and any changes in fluorescence intensity using Eq. (2). Finally mean ± SEM were estimated from four experiments and plotted.

Binding of proteins in the vicinity of the fluorescein moiety often leads to quenching because the fluorescence quantum yield of the

bound species is typically lower than that of free ssDNA. Unless corrected for, this artifact can result in significant errors in the estimation of binding affinity[42].

$$\frac{F_b}{F_f} = \left[\frac{A - A_f}{A_b - A}\right] \times \frac{Q_f}{Q_b} = \frac{A_c - A_f}{A_b - A_c}$$

Rearranging, we get

$$A_c = \frac{\left[\left(\frac{A-A_f}{A_b-A}\right) \cdot \left(\frac{Q_f}{Q_b}\right) \cdot (A_b)\right] + A_f}{1 + \left[\left(\frac{A-A_f}{A_b-A}\right) \cdot \left(\frac{Q_f}{Q_b}\right)\right]} \quad (2)$$

Where, (1) $F_b$, and $F_f$ are the bound, and free concentrations of the FAM-labeled fluorescent ssDNA in μM, (2) $Q_b$, and $Q_f$ are the fluorescence quantum yields of the bound and free form of the FAM-labeled fluorescent ssDNA (arbitrary units), (3) $A_b$, and $A_f$ are the anisotropy values of the bound, and free forms of the FAM-labeled fluorescent ssDNA, (4) $A$, is the measured anisotropy, and (5) $A_c$ is the corrected anisotropy value

$K_D$ was determined after fitting corrected anisotropy values ($A_c$;using Eq. (2)) with a model for one site specific binding with Hill slope as defined by Eq. (3) using GraphPad Prism 9.

$$A_c = \frac{A_{c;max}[RPA]^h}{K_D{}^h + [RPA]^h} \quad (3)$$

Where, $A_c$ is the corrected anisotropy value from Eq. (2), $A_{c;max}$ is the maximum anisotropy when 100% of ssDNA is complexed with RPA, $K_D$ is the apparent dissociation constant, [$RPA$] is the concentration RPA in the cuvette after each successive addition, and $h$ is the Hill coefficient.

For the anisotropy experiments where the binding density of RPA was measured, 30 nM 5′-FAM-(dT)$_{35}$ or 5′-FAM-(dT)$_{70}$ were taken in 1x RPA reaction buffer in a 10 mm pathlength quartz cuvette (Firefly Sci) with stirring. RPA alone or RPA + Rtt105 (1:1) were titrated and after an incubation period of 3 min fluorescence anisotropy was measured and plotted against the concentration of proteins. RPA and Rtt105 were mixed at 1:1 molar ratio (1.1 μM each) in 1x RPA reaction buffer and incubated on ice for 30 min prior to each experiment. Measured anisotropy values were corrected for the G-factor and any changes in fluorescence intensity using Eq. (2); mean ± sem were estimated from three experiments and plotted.

The saturation points were taken as the intersection of biphasic or triphasic curves from the linear fits of the initial data points reflecting the change in anisotropy upon binding of sub-saturating amounts of proteins. For (dT)$_{35}$, the first dotted line represents a stoichiometry of 1:1, and the other represents 2.7:1. For (dT)$_{70}$, first dotted line represents a stoichiometry of 2:1, and the other represents 4.7:1. However, note that stoichiometry estimates with (dT)$_{70}$ is probably an underestimate because an anisotropy value of 0.19 is close to the limiting value of anisotropy measurements for fluorescein.

### Crosslinking mass spectrometry (XL-MS) analysis
Stock solutions of Rtt105 (13.4 mg/mL) and RPA (1.77 mg/mL) were diluted to 0.89 mg/mL and 0.3752 mg/mL, respectively in buffer (30 mM HEPES, 200 mM KCl, pH 7.8 and incubated together for 30 min. The diluted proteins were reacted with 5 mM bis(sulphosuccinimidyl)suberate (BS3) and 20 μL of the sample was taken at various time points (0, 15 and 30 min) and immediately quenched with 2 μL of 1 M ammonium acetate. Quenched samples were diluted with 1.5X Laemmli gel loading buffer to a final volume of 40 μL, vortexed, and heated to 100 °C for 5 min and resolved on 4–20% (w/v) gradient SDS-PAGE gels (Bio-Rad) with Tris-glycine buffer. Gels were stained with Gelcode blue safe protein stain (Thermo Scientific). Gel bands were excised for protein identification and analysis.

Excised bands were destained with a 50 mM ammonium bicarbonate and 50% acetonitrile mixture and reduced with a mixture of 100 mM DTT and 25 mM ammonium bicarbonate for 30 min at 56 °C. The reaction was subsequently exchanged for the alkylation step with 55 mM iodoacetamide and 25 mM ammonium bicarbonate and incubated in the dark at room temperature for 25 min. The solution was then washed with the 50 mM ammonium bicarbonate and 50% acetonitrile mixture. The gel pieces were then first dehydrated with 100% acetonitrile and then rehydrated with sequence grade trypsin solution (0.6 μg, Promega) and incubated overnight at 37 °C. The reaction was quenched with 10 μL of 50% acetonitrile and 0.1% formic acid (FA, Sigma) and transferred to new microfuge tubes, vortexed for 5 min, and centrifuged at 16,000 g for 30 min. Samples were transferred to mass spectrometry vials and quantitated by LC-MS as described for peptide identification[43,44]. Peptides were identified as previously described[45] using MassHunter Qualitative Analysis, version 6.0 (Agilent Technologies), Peptide Analysis Worksheet (ProteoMetrics LLC), and PeptideShaker, version 1.16.42, paired with SearchGUI, version 3.3.16 (CompOmics). Crosslinks were then determined using Spectrum Identification Machine (SIMXL 1.5.5.2).

### Hydrogen-deuterium exchange mass spectrometry (HDX-MS) analysis
Stock solutions of RPA (13.4 mg/mL) and Rtt105(1.77 mg/mL) were mixed in the presence or absence of (dT)$_{35}$ ssDNA in a 1:1.2 ratio. Reactions were diluted 1:10 into deuterated reaction buffer (30 mM HEPES, 200 mM KCl, pH 7.8). Control samples were diluted into a non-deuterated reaction buffer. At each time point (0, 0.008, 0.05, 0.5, 3, 30 h), 10 μL of the reaction was removed and quenched by adding 60 μL of 0.75% formic acid (FA, Sigma) and 0.25 mg/mL porcine pepsin (Sigma) at pH 2.5 on ice. Each sample was digested for 2 min with vortexing every 30 s and flash-frozen in liquid nitrogen. Samples were stored in liquid nitrogen until the LC-MS analysis. LC-MS analysis of RPA was completed as described[46]. Briefly, the LC-MS analysis of RPA was completed on a 1290 UPLC series chromatography stack (Agilent Technologies) coupled with a 6538 UHD Accurate-Mass QTOF LC/MS mass spectrometer (Agilent Technologies). Peptides were separated on a reverse phase column (Phenomenex Onyx Monolithic C18 column, 100 × 2 mm) at 1 °C using a flow rate of 500 μl/min under the following conditions: 1.0 min, 5% B; 1.0 to 9.0 min, 5 to 45% B; 9.0 to 11.8 min, 45 to 95% B; 11.8 to 12.0 min, 5% B; solvent A = 0.1% FA (Sigma) in water (Thermo Fisher) and solvent B = 0.1% FA in acetonitrile (Thermo Fisher). Data were acquired at 2 Hz s$^{-1}$ over the scan range 50 to 1700 m/z in the positive mode. Electrospray settings were as follows: the nebulizer set to 3.7 bar, drying gas at 8.0 L/min, drying temperature at 350 °C, and capillary voltage at 3.5 kV. Peptides were identified as previously described[45] using MassHunter Qualitative Analysis, version 6.0 (Agilent Technologies), Peptide Analysis Worksheet (ProteoMetrics LLC), and PeptideShaker, version 1.16.42, paired with SearchGUI, version 3.3.16 (CompOmics). Deuterium uptake was deter- mined and manually confirmed using HDExaminer, version 2.5.1 (Sierra Analytics). Heat maps were created using MSTools[47].

### MB543 fluorescence quenching assay to estimate binding affinity between RPA and Rtt105
RPA-DBD-A$^{MB543}$, RPA-DBD-D$^{MB543}$, or F-A-B-DBD-A$^{MB543}$ were diluted to 200 nM in 1x RPA reaction buffer. 180 μl of either fluorescent protein was added to a 3 mm path length quartz cuvette (Starna Cells Inc.) and maintained at 23 °C in a PC1 spectrofluorometer (ISS Inc.). The MB543 dye, conjugated to either DBD-A or DBD-D domain of RPA, was excited at 535 nm and the resulting fluorescence emission spectra were collected between 558 nm to 578 nm with the $\lambda_{max}$ located at 568 nm. Unlabeled Rtt105, diluted to a 4 μM stock in RPA

reaction buffer, was added to the cuvette in a stepwise manner, mixed, and incubated for 3 min to achieve equilibrium before emission spectra were measured. Emission scans were recorded twice from at least two cuvettes, and the experiment was repeated three to four times. The fluorescence at $\lambda_{max}$ from 3–4 trials was corrected for stepwise dilution of sample (<8%), normalized to the initial fluorescence (fluorescence intensity in the absence of Rtt105), and plotted as mean and SEM. The fluorescence intensity values were transformed to fraction quenched versus Rtt105 concentration and fitted to a quadratic Eq. (4) using non-linear least squares regression, accounting for ligand depletion, to yield an apparent equilibrium constant ($K_D$). The dilution factor corrected RPA-DBD-D$^{MB543}$ fluorescence remained nearly constant (within error) and served as a control for change in fluorescence due to photobleaching alone.

$$i = i_{min} + (i_{max} - i_{min}) \times$$

$$\left( \frac{\left(K_D + [RPA^f] + [Rtt105]\right) - \sqrt{\left(K_D + [RPA^f] + [Rtt105]\right)^2 - 4[RPA^f][Rtt105]}}{2[RPA^f]} \right) \quad (4)$$

Where, $i$, is the measured fluorescence intensity, $i_{min}$, & $i_{max}$ are minimum and maximum values of the fluorescence intensity of 100% free, and 100% bound RPA determined from the fit, respectively. $[RPA^f]$, is the concentration of fluorescent RPA taken in the cuvette and the value is constrained at 200 nM. However, note that due to the stepwise addition of Rtt105 there is a 5% dilution by the end of the measurement. $[Rtt105]$ is the dilution factor corrected concentration of Rtt105 in the cuvette in nM, and $K_D$ is the dissociation constant determined from the fit.

## Measurement of DNA binding kinetics using stopped flow fluorescence
All stopped flow experiments were performed on a SX20 instrument (Applied Photophysics Inc.) at 25 °C in 1x RPA reaction buffer. Protein and or DNA reactions from individual syringes were rapidly mixed and fluorescence data were collected. The respective mixing schemes are denoted by cartoon schematics within the figure panels. Seven to eight individual shots were averaged for each experiment. All experiments were repeated a minimum of 3 times and SEM from the individual fits are noted in the figure legends. For the FRET experiments, samples were excited at 535 nm (Cy3 wavelength) and Cy5 emission was captured using a 645 nm long-pass filter. For the RPA-Rtt105 interactions, RPA-DNA and Rtt105-RPA-DNA interactions, experiments were performed with 100 nM each of RPA, Rtt105, and (dT)$_{35}$ or (dT)$_{40}$ ssDNA substrates (1:1:1 ratio). For facilitated exchange stopped flow experiments, 200 nM RPA-DBD-A$^{Cy5}$ and 200 nM RPA-DBD-D$^{Cy3}$ were premixed with 120 nM (dT)$_{97}$ and shot against unlabeled RPA (500 nM) or the RPA-Rtt105 complex (500 nM each).

## Steady State Förster resonance energy transfer (FRET) measurement of protein-DNA and protein-protein complexes
RPA-DBD-D$^{Cy3}$ and Rtt105$^{Cy5}$ were mixed in 1x RPA reaction buffer at 1:1 ratio such that the final concentration of each protein was 200 nM. The complex was incubated on ice for 30 min and then transferred to a 3 mm pathlength cuvette maintained at 23 °C in the PC1 spectrofluorometer. The sample was excited at 535 nM and the resulting fluorescence between 550 nm to 700 nm was collected as a FRET spectrum. Next, ssDNA of different lengths (dT$_x$); where x = (dT)$_8$, (dT)$_{15}$, (dT)$_{25}$, (dT)$_{35}$, (dT)$_{45}$, (dT)$_{54}$, (dT)$_{64}$, (dT)$_{70}$, or (dT)$_{84}$ were added in a stepwise manner to the cuvette, mixed, and incubated for 3 min at 23 °C before FRET spectra was again recorded after each addition. Raw spectra were corrected by incorporating the estimated dilution factor and then area normalized to account

for any fluctuations in lamp intensity. Finally, ratiometric FRET was calculated as defined by Eq. (5):

$$FRET = \frac{I_A}{I_A + I_D} \quad (5)$$

Where, $I_A$ and $I_D$ are acceptor and donor fluorescence emission intensities at respective $\lambda_{max}$ = 673, and 573 nM, respectively.

Quantitative FRET between RPA-DBD-D$^{Cy5}$ or -A$^{Cy5}$ and 5′ or 3′ labelled Cy3-(dT)$_{40}$, respectively was estimated by correcting for the increase in acceptor fluorescence during titration. This monotonic increase was estimated and subtracted from FRET spectra. In addition, protein induced fluorescence enhancement or PIFE, was estimated by titrating in -D$^{Cy5}$ or -A$^{Cy5}$ with 3′ or 5′ labelled Cy3-(dT)$_{40}$. This was also subtracted to estimate true energy transfer values.

## C-Trap optical tweezer analysis of RPA-Rtt105 interactions
48.5 kbp Lambda DNA construct was prepared with three biotins on either end of DNA. Lambda DNA was purchased from Roche Inc. Short oligos to anneal to the sticky ends were purchased from Integrated DNA Technologies Inc. DNA were stored in TE buffer (10 mM Tris-HCl, pH 8.0 and 0.1 mM EDTA). Optical trap experiments were performed using a commercial dual optical trap combined with confocal microscopy and microfluidics [C-trap] from Lumicks BV Inc. Streptavidin coated polystyrene particle beads of average size 4.8 µM [0.5% w/v] (Spherotech Inc.) were diluted 1:250 in 1X PBS and 1–2 nM of DNA were made in 1X PBS. DNA was captured between two streptavidin beads and mechanically denatured by moving one bead to create ssDNA. ssDNA was confirmed by fitting force-distance (FD) curve to Freely Jointed Chain model [FJC] (contour length 48.5 kbp/16.49 µm; persistence length 46 nm; stretch modulus 1000 pN) in real time. DNA was held for 5 s in the fully ssDNA state and then returned to 5 pN tension for the fluorescence experiments. RPA-DBD-D$^{MB543}$ in storage buffer (30 mM HEPES pH 7.8, 200 mM KCl, 0.02% Tween-20, 10% glycerol, and 0.2 mM EDTA) was added to the DNA in the absence or presence of Rtt105$^{Cy5}$. Rtt105$^{Cy5}$ was kept in storage buffer with 1 mm TCEP-HCl. Both proteins were diluted to 1 nM with experimental buffer (30 mM HEPES pH 7.8, 100 mM KCl, 6% Glycerol, 5 mM MgCl$_2$ and incubated together (for the experiments where Rtt105-RPA complexes were tested) at 1:1 molar ratio (10 pM final concentration each). Imaging buffer 0.8% (w/v) dextrose, 165 U/mL glucose oxidase, 2170 U/mL catalase, and 2–3 mM Trolox was used to increase the fluorescence lifetime of the fluorophores. Imaging settings were 2–3 ms exposure time (per pixel), red excitation 638 nm, and green excitation 561 nm. Data was analyzed using custom python script [Pylake API from Lumicks].

## Analytic ultracentrifugation (AUC) analysis
AUC sedimentation velocity experiments were performed on an Optima analytical ultracentrifuge (Beckman-Coulter Inc.) using an An-50Ti rotor at 40,000 rpm at 20 °C. Proteins and DNA either alone or in complex were dialyzed against 30 mM HEPES, pH 7.8, 100 mM KCl, 10% glycerol, and 1 mM TCEP-HCl before each experiment. Concentrations used for the experiments are mentioned in the appropriate figures. Sample (380 µL) and buffer (400 µL) were filled in each chamber of a 2-sector charcoal quartz cell. Absorbance was monitored at 280 nm and/or 650 nm. Since the absorbance signal from Rtt105 was low, Rtt105$^{Cy5}$ was used and tracked at 650 nm. Scans were recorded at 3 min intervals. The density and viscosity of the buffer at 20 °C were calculated using SEDNTERP. Continuous distribution (c(s)) model was used to fit the data in SEDFIT[48].

## Reporting summary
Further information on research design is available in the Nature Research Reporting Summary linked to this article.

## Data availability

All data are available within this manuscript and it supplementary information files. Plasmids used for protein overexpression and the data supporting the findings of this study are available from the corresponding author upon request. Source data are provided with this paper.

## Code availability

Code for C-trap data analysis is available at https://github.com/spangeni/Rtt105-Analysis-Pipeline/tree/main.

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

## Acknowledgements

The authors thank members of our respective research laboratories for critical reading of the manuscript. This work was supported by grants from the National Institutes of Health (R01 GM130746 and R01 GM133967) to E.A., (R15 GM123443) to H.B., (R35 GM122569) to T.J. and (R01 GM131058 and R35 GM145260) to X.Z. Funding for Proteomics, Metabolomics and Mass Spectrometry Facility at MSU was made possible in part by the MJ Murdock Charitable Trust and NIGMS of the National Institutes of Health under Award Number P20 GM103474. The analytical ultracentrifuge experiments and C-trap experiments were supported by instrumentation grants from the Office of the Director, National Institutes of Health (S10 OD030343 to E.A. and S10 OD025221 to Johns Hopkins University, respectively). M.K.S. was supported by a summer undergraduate research supplement from the National Institutes of Health (R01 GM130746-04S1 to E.A.).

## Author contributions

S.K., J.D., R.C., N.P., V.K. and M.K.S. purified proteins, generated fluorescent protein, designed, and performed the steady-state and pre-steady state experiments. J.M., A.P., S.K., and B.B. performed and analyzed the HDX-MS and XLMS experiments. S.P. and T.H. performed and analyzed the C-trap experiments. N.D. and X.Z. performed and analyzed the in vivo experiments. S.S. and H.B. performed and analyzed single molecule FRET measurements of GQ-unwinding. E.A. directed the project, designed experiments, and composed the manuscript. All authors contributed to manuscript preparation.

## Competing interests

The authors declare no competing interests.
