## [Peer Review File · Nature Communications]

Rtt105 regulates RPA function by configurationally stapling the flexible domainsEditorial Note: Parts of this Peer Review File have been redacted as indicated to maintain the confidentiality of unpublished data.

REVIEWER COMMENTS

Reviewer #1 (Remarks to the Author):

The manuscript by Kuppa et al. investigates how Rtt105 engages with and regulates RPA interactions with other proteins and single-stranded DNA. The authors use a combination of structural, biochemical and single-molecule approaches to demonstrate that Rtt105 engages and conformationally restricts several DNA binding and protein-interaction domains within the N-terminal region. Interestingly, they find that Rtt105 binding to RPA inhibits engagement by Rad52, however, after helping RPA organize filaments on long ssDNAs, Rtt105 is released and can no longer engage with high affinity. Finally, the authors performed genetic experiments to demonstrate the functional distinctions between Rtt105 binding and ssDNA binding activity.

Understanding the molecule pathways by which RPA-interacting proteins bind, conformationally reorganize, and regulate RPA is a timely topic of broad interest due to the critical role RPA plays in a wide array of pathways. The manuscript presents a true tour de force of methods and weaves the various findings together into a very convincing narrative. The manuscript was a pleasure to read and puzzle through with realizations from one approach nicely linked to observations in the next. Given that many of the central conclusions from the work revolved about the authors configurational stapling model, it would have been nice if the structure was solvable by cryo-em, but clearly the intrinsic dynamic nature of RPA and Rtt105 make this difficult, if not impossible. Nevertheless, I found the collection of approaches used to gather structural information very convincing and a great illustration of how these types of dynamic assemblies can be studied.

Overall, the manuscript is well-written, the experiments were well controlled and clearly presented in the figures, but I was left with a few lingering technical questions that I feel should be addressed (my specific comments below). Also, in the title the authors state that Rtt105 configurational stapling of RPA blocks interactions with RPA-interacting proteins. I feel this is too general of a statement since the authors only evaluated Rad52 in the manuscript. More on this in my comments below.

Comments:

The significance of the fluorescent amplitude signal differences in Figure 2 h and j are not clearly explained in the main text (top of page 9). I agree the binding kinetics clearly show a large difference, but how do the authors interpret the difference in amplitude. If the difference is due to a real change in distance, equilibrium FRET measurements should show the same difference. Do the authors see a similar difference for equilibrium measurements?

At the bottom of page 9 the authors state that the stopped flow experiments reveal configurational change in RPA. Are the authors suggesting the conformation of RPA changes on the ssDNA during binding? How can this be concluded from the stopped flow measurements? The changes in amplitude could suggest different conformations bound to Rtt105 prior to engagement with ssDNA that result in a different final structure when bound to ssDNA. Can the authors exclude this possibility?

In the manuscript, the authors suggest that the inhibition of Rad52 by the Rtt105 interaction with RPA is reflective of a general feature of how one RPA-interacting protein could inhibit another. They suggest a main function of Rtt105 binding to RPA is then to inhibit and regulate interactions with other RIPs. The authors should discuss why it is reasonable to conclude this is a general property based only on experiments using Rad52. The current statement in the title is too broad in the absence of additional experimental support or a more convincing explanation. What are the structural similarities and differences between Rad52 and other RIPs? Are these consistent with the possibility of a broader trend?

Minor comments:

On page 5 there appears to be a typo. It says "an F-A-B version of RPA containing just OB-F, DBD-A and DBD-D interacts with Rtt105." The F-A-B should be subunits OB-F, DBD-A and DBD-B since this is what is stated elsewhere and DBD-D was shown on to interact.

Data is plural and should be referred to throughout. On page 5 toward the bottom "The data" should be changed to "These data."

The title is very wordy and confusing. It would be improved if it were focused a bit. Removing one of the and clauses would help a lot.

Reviewer #2 (Remarks to the Author):

The manuscript reports a study that investigates the interaction between RPA and Rtt105, and also evaluates the role of single-stranded DNA in the interactions. By using XL-MS and HDX-MS as tools, the authors suggested a model where Rtt105 binds to several DNA-binding and protein-binding domains and further stabilizes the complex. Overall, the topic is of broad interest and fits in the scope of Nature Communications and can be considered for publication after addressing these concerns listed below.

1. In page 5, 7th line from the bottom, authors wrote "better affinity", which should be corrected as "higher affinity".

2. In page 6, Figure 1, authors show the structural model for the different RPA subunits. Are these structures resolved from high-resolution techniques and deposited in PDB? If so, the authors should cite corresponding literatures and PDB IDs. If not, the authors should explain how these structural models were obtained.

3. In page 12, line 7, authors mentioned that they achieved "excellent peptide coverage". Given the coverage lies between 67% to 92%, excellent is clearly an overstatement here. Moreover, when referring to Figure S4, there is an obvious mis-match between the number reported in the manuscript and the coverage map shown in Figure S4. With such sequence coverage (e.g., 30% for RPA32 in Figure S4), it's challenging to obtain a good-enough understanding to propose any structural models.

4. Is the sequence coverage a sum of both HDX and XL-MS? Or is it just for HDX? There are a number of reasons that can lead to low sequence coverage in HDX, including but not limited to glycosylation, limited pepsin digestion sites, less optimized digestion conditions, less optimized protein denaturation/quench conditions. From Figure S4, the figure resolution is not sufficient to tell the protein sequence, so that I cannot reach to a conclusion based upon current information. In any event, authors should comment on the sequence coverage and also perform further experiments if necessary to increase the sequence coverage.

5. In page 5, end of 2nd paragraph, authors indicate that more crosslinks are observed between RPA70 and Rtt105 as compared with RPA32-Rtt105 and RPA14-Rtt105 complexes, and this is in agreement with their prior observation related to binding affinity. Number of cross-links is determined by many contributing factors, and higher number of cross-links does not necessarily indicate a higher binding affinity. Same logic applies to HDX data interpretation, where higher Δ HDX is not directly related with higher binding affinity.

6. As authors obtained extensive structural information with the help of XL-MS and HDX-MS, authors may consider conducting a docking exercise using restraints from XL-MS (possible distances between two cross-linked alpha carbons) and HDX (regions that are either exposed or buried upon binding).

This will not only help the audience better visualize the structural model that the authors are trying to propose, but also help provide additional support to the proposed model. There have been previous reports utilizing this strategy for structural modeling (e.g., *Anal. Chem.* 2019, 91, 24, 15709–15717).

7. Another minor concern about HDX results is about the kinetic curves shown in supplemental information. For example, in Figure S8, peptide 291-302, the deuterium uptake for the Rtt105 bound state decreases from 0.008 to 0.05h. This is abnormal, but given the error bar, it seems to be confident. Can the authors comment on this observation, maybe it's related to experiment execution as 0.008h is fairly short?

8. By reading the experimental conditions in page 28, authors may consider adding denaturant into the quench solution and utilize high-pressure online pepsin digestion to further increase the HDX coverage.

Reviewer #3 (Remarks to the Author):

The manuscript "Rtt105 configurationally staples RPA and blocks facilitated exchange and interactions with RPA-interacting proteins" by Kuppa et al. presents an extensive analysis on the interactions between Replication Protein A and the molecular chaperone Rtt105 in presence and absence of ssDNA. Further, the authors tested the interactions of the complex with the mediator of homologous recombination, Rad52.

For me as a reader not familiar with RPA biology, the information density of the manuscript is almost too high. A myriad of technologies were applied (SDS page gels, stopped flow, FRET, C-trap, crosslinking mass spectrometry, analytical ultracentrifugation analysis,...) making reading and understanding the manuscript very challenging. Having said that, the experiments are well designed and well executed enabling the authors to draw strong and supported conclusions. Overall, I consider the manuscript a very valuable addition the field and I will focus my (few) minor comments mainly on improving accessibility and clarity of the manuscript.

Some specific comments

1. I suggest to mention already in the abstract that RPA is a heterotrimeric complex.
2. (Figure 2 and others). To better indicate the emission wavelengths of the dyes, I suggest to colour the acceptor with a pink/red star and the donor with a blue one
3. (Figure 2e). Please plot the (two) data points per condition rather than a box plot
4. Figures S8-S12 were missing in the pdf I got
5. (Figure 4) e) and f) should read d) and e)
6. (page 15) Suggestion to replace RPA molecules with RPA complexes
7. (figure 6). In d), the authors monitor disappearance of RPA over a time span of ~60s, in e), the ssDNA seems to rupture after around 30s. Maybe the authors made another run, where the length of observation is comparable?
8. (Figure 7a): Units on the x-axis are missing.
9. (page 31) 535 nm not 535 nM

Reviewer #4 (Remarks to the Author):

Kuppa and colleagues investigated the role of Rtt105 in regulating RPA interactions with ssDNA and Rad52. They carried out a large array of rigorous and systematic bulk biochemical, in vivo, and single-molecule experiments. These data show that Rtt105 binds to multiple DNA binding and protein-interaction domains of RPA, and these interactions are dynamic and regulate interactions with RPA interactions with other proteins depending on the availability of ssDNA. I appreciate the diverse range of biochemical, biophysical, and in vivo work presented in this manuscript, and some of the findings are potentially significant. That being said, I also struggle with the coherence of the findings and how to properly place this work in the context of existing literature.

1. The authors stated that RPA-Rtt105 interactions and RPA-ssDNA binding are functionally distinct, and RPA has a comparable binding affinity (KD) to ssDNA with and without Rtt105 present. On the other hand, they also show that Rtt105 influences the ssDNA binding kinetics and configurations of RPA, with extensive contacts between Rtt105 and DBDs of RPA. While the latter indicates that Rtt105 and ssDNA compete for binding to RPA and have significant crosstalk in their RPA binding interactions, the former implies that RPA-Rtt105 interactions and RPA-ssDNA binding function independently. There is an apparent contradiction, which is not clearly explained. Is the finding that RPA has a comparable binding affinity to ssDNA with and without Rtt105 a mere coincidence?

2. On page 6, "RPA binds stoichiometrically to a 5'-Cy5-(dT)30 oligonucleotide in electrophoretic mobility band-shift analysis (EMSA) and a preformed Rtt105-RPA complex does not influence the ssDNA binding activity of RPA (KD=5.5±1.1 and 6.7±2.1 nM for RPA and RPA-Rtt105, respectively; Figures 2a & b)". Based on this statement, KD for RPA is smaller than KD for RPA-Rtt105, but the raw data points (compare grey and blue data points) suggest an opposite trend in Figure 2b. Is this a typo?

3. Figures 2h, 3h, 6a, and 6b show a transition in FRET noise level at 1s, likely due to a change in data filtering. The authors need to explain this transition. Whenever possible, it would be more helpful to use the same filtering for a given data set.

4. The story might be easier to follow if Figure 4, which shows the contact maps of Rtt105 and RPA, were presented as Figure 1.

5. The abstract places much emphasis on Rad52, but the focus on Rad52 in the main text is limited.

Response to reviewer comments

We thank all four reviewers for their time and valuable advice. Their words of support and encouragement are much appreciated. Our comments to the reviewer suggestions are denoted in blue.

Reviewer #1 (Remarks to the Author)

The manuscript by Kuppa et al. investigates how Rtt105 engages with and regulates RPA interactions with other proteins and single-stranded DNA. The authors use a combination of structural, biochemical, and single-molecule approaches to demonstrate that Rtt105 engages and conformationally restricts several DNA binding and protein-interaction domains within the N-terminal region. Interestingly, they find that Rtt105 binding to RPA inhibits engagement by Rad52, however, after helping RPA organize filaments on long ssDNAs, Rtt105 is released and can no longer engage with high affinity. Finally, the authors performed genetic experiments to demonstrate the functional distinctions between Rtt105 binding and ssDNA binding activity.

Understanding the molecule pathways by which RPA-interacting proteins bind, conformationally reorganize, and regulate RPA is a timely topic of broad interest due to the critical role RPA plays in a wide array of pathways. The manuscript presents a true tour de force of methods and weaves the various findings together into a very convincing narrative. The manuscript was a pleasure to read and puzzle through with realizations from one approach nicely linked to observations in the next. Given that many of the central conclusions from the work revolved about the authors' configurational stapling model, it would have been nice if the structure was solvable by cryo-em, but clearly the intrinsic dynamic nature of RPA and Rtt105 make this difficult, if not impossible. Nevertheless, I found the collection of approaches used to gather structural information very convincing and a great illustration of how these types of dynamic assemblies can be studied.

Overall, the manuscript is well-written, the experiments were well controlled and clearly presented in the figures, but I was left with a few lingering technical questions that I feel should be addressed (my specific comments below). Also, in the title the authors state that Rtt105 configurational stapling of RPA blocks interactions with RPA-interacting proteins. I feel this is too general of a statement since the authors only evaluated Rad52 in the manuscript. More on this in my comments below.

- We thank the reviewer for the supportive comments. Specific answers to the suggestions/questions are appended below. We do agree with the suggestion for the title and have changed it as recommended. The new title reads: "Rtt105 regulates RPA by configurational stapling of flexible domains".

Comments:

1. The significance of the fluorescent amplitude signal differences in Figure 2 h and j are not clearly explained in the main text (top of page 9). I agree the binding kinetics clearly show a large difference, but how do the authors interpret the difference in amplitude. If the difference is due to a real change in distance, equilibrium FRET measurements should show the same difference. Do the authors see a similar difference for equilibrium measurements?

- This is the text in the manuscript "*These data show that DBD-A binds with ~similar rates to the 5' end-labeled DNA in the absence or presence of Rtt105 (Figures 2g & h). However, the fluorescence signal amplitude in the presence of Rtt105 is twice that observed for RPA alone (Figure 2h). In contrast, Rtt105 reduces the rate of DBD-D binding to the 3' end-labeled DNA (Figures 2i & j), while the signal amplitudes remain similar (Figure 2j). These data suggest that*

Rtt105 differentially influences the two DBDs of RPA and support a model where Rtt105 could drive formation of specific configurations RPA (described below)."

As the reviewer mentioned, in pre-steady state kinetic experiments the first 10 seconds of encounter between RPA, and ssDNA molecules (\pm Rtt105) is monitored using FRET as a measure of molecular proximity. We note that there is a change in the amplitude when the FRET pair is positioned between DBD-A and the 5' end of the DNA, and not between DBD-D and the 3' end. In the pre-steady state experiments, we are looking at the rearrangements of these domains as either RPA or the RPA-Rtt105 complex first encounters ssDNA. As shown in the paper, RPA (+/Rtt105) will bind to ssDNA and undergo rearrangements/diffusion/remodeling and Rtt105 dissociates under conditions when more than one RPA binds to DNA. In fact, on a (dT)₄₀ substrate, multiple RPA molecules can bind as the ratio of RPA is increased (analytical ultracentrifugation analysis shown below). Thus, at equilibrium, we predict an ensemble of configurational states with multiple RPA bound on DNA in solution. In such steady state experiments we expect signal changes arising from FRET to be minimal considering dynamic RPA molecules on ssDNA. We performed the experiment suggested by the reviewer, and we precisely see this behavior (Figure below).

2. At the bottom of page 9 the authors state that the stopped flow experiments reveal configurational change in RPA. Are the authors suggesting the conformation of RPA changes on the ssDNA during binding? How can this be concluded from the stopped flow measurements? The changes in amplitude could suggest different conformations bound to Rtt105 prior to engagement with ssDNA that result in a different final structure when bound to ssDNA. Can the authors exclude this possibility?

- Yes, the reviewer is correct. We cannot decipher the individual contributions of ssDNA and Rtt105 to the observed changes in overall FRET.

Here is the text in the manuscript: *“These data show that the configurational changes in RPA observed in the stopped flow experiments (Figure 2) are induced while both Rtt105 and DNA are simultaneously bound to RPA. We propose that the RPA-Rtt105 complex binds ssDNA and both proteins are likely reconfigured as the DBDs engage onto ssDNA.”*

We are merely suggesting that both Rtt105 and RPA are present as a transient complex on ssDNA when these fluorescence changes occur; i.e., the lifetime of the hetero-trimeric Rtt105-RPA-ssDNA complex is at least 10s (or the time duration of the pre-steady state kinetic experiments). The likelihood of Rtt105 leaving the complex increases as more RPA molecules bind the DNA. We want to reinforce the finding that Rtt105 does not readily leave the RPA-ssDNA complex on short ssDNA substrates where RPA and DNA are present in 1:1 molar ratios. We certainly expect Rtt105 to drive certain configurations (as shown in Figure 3) and ssDNA to remodel this complex. We summarized the interpretation with this statement in the manuscript: *“Thus, the interactions between RPA and Rtt105 are different in the DNA bound versus unbound states.”*

3. In the manuscript, the authors suggest that the inhibition of Rad52 by the Rtt105 interaction with RPA is reflective of a general feature of how one RPA-interacting protein could inhibit another. They suggest a main function of Rtt105 binding to RPA is then to inhibit and regulate interactions with other RIPs. The authors should discuss why it is reasonable to conclude this is a general property based only on experiments using Rad52. The current statement in the title is too broad in the absence of additional experimental support or a more convincing explanation. What are the structural similarities and differences between Rad52 and other RIPs? Are these consistent with the possibility of a broader trend?

- We changed the title of the paper to reflect this point raised by the reviewer. For both yeast and RPA, we have been tracking the published information about protein-protein interactions. Unfortunately, most such interactions have only been established using peptides of the RPA-interacting proteins (RIPs) and either the winged-helix domain (PID^{32C}) or OB-F (PID^{70N}). For human RPA binding to DNA2, this complex has been shown to form in the presence of DNA. Thus, tracking down a well characterized RPA-RIP interaction using full-length proteins has been a challenge. We purified yeast Fen1 and tested binding with RPA and we are not able to capture that interaction under our solution conditions. This exploration essentially reveals the void in information about RPA-RIP interactions and we will be focusing on exploring and understanding these interactions using more rigorous biophysical approaches.

Our work on yeast RPA-Rad52 interactions also do not match with previously published work. We can reliably capture RPA-Rad52 binding in the absence of ssDNA and now also have a preliminary cryoEM map of this complex (unpublished work shown in Figure below). This complex is thought to be formed between the disordered C-terminus of Rad52 and two binding sites in RPA (one in PID^{32C}, and the other at an undefined position between DBDs-A and B). However, in the cryoEM density, RPA is bound to the ordered part of Rad52, and deletion of the disordered region ablates RPA binding. Thus, there appears to be extensive allostery involved in these interactions.

At a minimum, given our rigorous investigation of the RPA-Rad52 complex, we are comfortable interpreting the RPA-Rad52 interaction and can reliably report the influence of Rtt105 on this complex. However, we would prefer to interpret other RPA-RIP interactions once we can accurately capture and characterize them.

We have expanded a bit more on the nature of the RPA-Rad52 complex in the discussion to shed light on our understanding of RPA-RIP interactions.

[redacted]

Minor comments:

4. On page 5 there appears to be a typo. It says “an F-A-B version of RPA containing just OB-F, DBD-A and DBD-D interacts with Rtt105.” The F-A-B should be subunits OB-F, DBD-A and DBD-B since this is what is stated elsewhere and DBD-D was shown on to interact.

- Thanks for catching the error. This has been corrected.

4. Data is plural and should be referred to throughout. On page 5 toward the bottom “The data” should be changed to “These data.”

- This has been corrected.

6. The title is very wordy and confusing. It would be improved if it were focused a bit. Removing one of the and clauses would help a lot.

- The new title reads: “Rtt105 regulates RPA function by configurationally stapling the flexible domains”.

Reviewer #2 (Remarks to the Author)

The manuscript reports a study that investigates the interaction between RPA and Rtt105, and also evaluates the role of single-stranded DNA in the interactions. By using XL-MS and HDX-MS as tools, the authors suggested a model where Rtt105 binds to several DNA-binding and protein-binding domains and further stabilizes the complex. Overall, the topic is of broad interest and fits in the scope of Nature Communications and can be considered for publication after addressing these concerns listed below.

1. In page 5, 7th line from the bottom, authors wrote “better affinity”, which should be corrected as “higher affinity”.

- Corrected.

2. In page 6, Figure 1, authors show the structural model for the different RPA subunits. Are these structures resolved from high-resolution techniques and deposited in PDB? If so, the authors should cite corresponding literatures and PDB IDs. If not, the authors should explain how these structural models were obtained.

- The following text has been added to the legend for Figure 1. “DBDs-C, D and E are from the cryoEM structure of *S. cerevisiae* RPA (PDB: 6I52). Homology models for DBD-F (PDB: 5N8A), DBD-A & DBD-B (PDB: 1JMC), and winged helix domains (PDB:4OU0) were built with Swiss-Model using the listed PDB files as template. DNA from 1JMC is docked onto DBD-A and B homology models.”

Corresponding papers are referred to in the manuscript.

3. In page 12, line 7, authors mentioned that they achieved “excellent peptide coverage”. Given the coverage lies between 67% to 92%, excellent is clearly an overstatement here. Moreover, when referring to Figure S4, there is an obvious mismatch between the number reported in the manuscript and the coverage map shown in Figure S4. With such sequence coverage (e.g., 30% for RPA32 in Figure S4), it’s challenging to obtain a good-enough understanding to propose any structural models.

-We agree with the reviewer that the coverage for all proteins should not be described as excellent. We changed this to read “good coverage” The comment pointed out above was intended to be specific to RPA70. The discrepancy between the text and Fig S4 has now been corrected. Given that the combined amino acid sequence of the four proteins under investigation is ~1200 residues, we are very satisfied with the coverage that was obtained. 1200 amino acids far exceeds the number of residues investigated in nearly all HDX experiments. With very large proteins, or multi-protein complexes, unequivocal identification of peptides is a challenge. We have only reported data for peptides which we are certain of the identification. Most importantly, we do have coverage in the key areas that are relevant to the proposed structural models.

4. Is the sequence coverage a sum of both HDX and XL-MS? Or is it just for HDX? There are a number of reasons that can lead to low sequence coverage in HDX, including but not limited to glycosylation, limited pepsin digestion sites, less optimized digestion conditions, less optimized protein denaturation/quench conditions. From Figure S4, the figure resolution is not sufficient to tell the protein sequence, so that I cannot reach to a conclusion based upon current information. In any event, authors should comment on the sequence coverage and also perform further experiments if necessary to increase the sequence coverage.

-We apologize for not being clear on the nature of the data presented in Figure S4. The sequence coverage shown in Figure S4 is for the HDX alone and does not include the XL-MS. Please see our reply to the comment above for a partial explanation of the coverage. An additional challenge, beyond the number of residues in the complex, was finding solution conditions in which all proteins and proteins/DNA complexes were highly soluble and showed no aggregation over time. This required us to work with concentrations well below standard HDX experiments. As explained above, this was done to ensure our data is of the highest quality and is without artifacts. By using orthogonal experiments (HDX, XL-MS, and fluorescence), we were able to corroborate our findings and to make sure our structure/function models were consistent with all data.

-We have improved the figure resolution and font size in Figure S4.

5. In page 5, end of 2nd paragraph, authors indicate that more crosslinks are observed between RPA70 and Rtt105 as compared with RPA32-Rtt105 and RPA14-Rtt105 complexes, and this is in agreement with their prior observation related to binding affinity. Number of cross-links is determined by many contributing factors, and higher number of cross-links does not necessarily

indicate a higher binding affinity. Same logic applies to HDX data interpretation, where higher Δ HDX is not directly related with higher binding affinity.

-We agree with the reviewer that the presence of cross-links cannot directly report on affinity. Use of the word “higher” in the final sentence of that paragraph was a poor choice. The sentence has now reads: “*These data agree with our observation that Rtt105 interacts more closely with the F-A-B half of RPA (Figure 1e)*”.

6. As authors obtained extensive structural information with the help of XL-MS and HDX-MS, authors may consider conducting a docking exercise using restraints from XL-MS (possible distances between two cross-linked alpha carbons) and HDX (regions that are either exposed or buried upon binding). This will not only help the audience better visualize the structural model that the authors are trying to propose, but also help provide additional support to the proposed model. There have been previous reports utilizing this strategy for structural modeling (e.g., Anal. Chem. 2019, 91, 24, 15709–15717).

-We agree with the reviewer that modeling the complex using computational tools is a good idea. This is an approach that we have used in the past to great effect (Ledbetter, Biochemistry 2017 Aug 15;56(32):4177-4190 and a review from our group – Tokmina-Lukaszewska Front Microbiol. 2018; 9: 1397.). However, there are caveats to docking and it does not always provide satisfactory results, particularly with proteins that have multiple conformations. Taking the prompt of the reviewer, we tried using RosettaDock as well as ClusPro 2.0. Neither approach gave results that confidently add to the data already presented. We will continue to pursue this approach as improved structural models for the domains become available and the resolution of our biophysical approaches improve.

7. Another minor concern about HDX results is about the kinetic curves shown in supplemental information. For example, in Figure S8, peptide 291-302, the deuterium uptake for the Rtt105 bound state decreases from 0.008 to 0.05h. This is abnormal, but given the error bar, it seems to be confident. Can the authors comment on this observation, maybe it’s related to experiment execution as 0.008h is fairly short?

-We thank the reviewer for the attention to detail. The amount of exchange cannot decrease over time in our experiments. While we performed all of the experiments in triplicate, as suggested by the reviewer there can be significant error at early time points and when exchange is low. We attribute the decrease shown in the data to this. While we have reported these peptides, we do not make any conclusions from this data.

8. By reading the experimental conditions in page 28, authors may consider adding denaturant into the quench solution and utilize high-pressure online pepsin digestion to further increase the HDX coverage.

-Thank you for these suggestions. This will be helpful, and we will try this in our ongoing experiments.

Reviewer #3 (Remarks to the Author)

The manuscript “Rtt105 configurationally staples RPA and blocks facilitated exchange and interactions with RPA-interacting proteins” by Kuppa et al. presents an extensive analysis on the interactions between Replication Protein A and the molecular chaperone Rtt105 in presence and absence of ssDNA. Further, the authors tested the interactions of the complex with the mediator of homologous recombination, Rad52.

For me as a reader not familiar with RPA biology, the information density of the manuscript is almost too high. A myriad of technologies were applied (SDS page gels, stopped flow, FRET, C-trap, crosslinking mass spectrometry, analytical ultracentrifugation analysis,...) making reading and understanding the manuscript very challenging. Having said that, the experiments are well designed and well executed enabling the authors to draw strong and supported conclusions. Overall, I consider the manuscript a very valuable addition to the field and I will focus my (few) minor comments mainly on improving accessibility and clarity of the manuscript.

Some specific comments

1. I suggest to mention already in the abstract that RPA is a heterotrimeric complex.
- We included this information in the first line of the abstract. "*Replication Protein A (RPA) is a heterotrimeric complex that binds to single-stranded DNA (ssDNA) and recruits over three dozen RPA-interacting proteins (RIPs) to coordinate multiple aspects of DNA metabolism including DNA replication, repair, and recombination.*"
2. (Figure 2 and others). To better indicate the emission wavelengths of the dyes, I suggest to colour the acceptor with a pink/red star and the donor with a blue one.
- The donor in our FRET experiments is Cy3 and colored pink in our schematics. The Cy5 is the acceptor and colored blue. This color scheme is maintained throughout the study. These colors reflect the corresponding natural colors (appearance) of the two fluorophores used and thus an intuitive interpretation for the readers.
3. (Figure 2e). Please plot the (two) data points per condition rather than a box plot.
- This has been changed as requested.
4. Figures S8-S12 were missing in the pdf I got.
- We have double checked these panels and they are present in our PDF copy. We would be happy to directly email the ms Word versions to the editor to be routed to the reviewer if needed.
5. (Figure 4) e) and f) should read d) and e)
- Thank you for catching the error. The mistake has been corrected.
6. (page 15) Suggestion to replace RPA molecules with RPA complexes
- The heterotrimer is constitutively held together and the subunits of RPA do not dissociate. Thus, in the context of the sentence, we would prefer to use 'molecules' as stoichiometry is being discussed.
7. (figure 6). In d), the authors monitor disappearance of RPA over a time span of ~60s, in e), the ssDNA seems to rupture after around 30s. Maybe the authors made another run, where the length of observation is comparable?
- In the representative data shown in Figure 6 c-e, ssDNA snaps around 50s. The black block between 5-15 seconds does not reflect breakage of ssDNA, but the fluorescence is turned off as the DNA bound bead is moved to a flow chamber containing either unlabeled RPA (Fig. 6d) or the RPA+Rtt105 complex (Fig. 6e). These are highlighted better in the Figure below.
- Images from a second molecule are also shown in the Figures below.

8. (Figure 7a): Units on the x-axis are missing.

- Corrected

9. (page 31) 535 nm not 535 nM

- Corrected

Reviewer #4 (Remarks to the Author)

Kuppa and colleagues investigated the role of Rtt105 in regulating RPA interactions with ssDNA and Rad52. They carried out a large array of rigorous and systematic bulk biochemical, in vivo,

and single-molecule experiments. These data show that Rtt105 binds to multiple DNA binding and protein-interaction domains of RPA, and these interactions are dynamic and regulate interactions with RPA interactions with other proteins depending on the availability of ssDNA. I appreciate the diverse range of biochemical, biophysical, and in vivo work presented in this manuscript, and some of the findings are potentially significant. That being said, I also struggle with the coherence of the findings and how to properly place this work in the context of existing literature.

1. The authors stated that RPA-Rtt105 interactions and RPA-ssDNA binding are functionally distinct, and RPA has a comparable binding affinity (K_D) to ssDNA with and without Rtt105 present. On the other hand, they also show that Rtt105 influences the ssDNA binding kinetics and configurations of RPA, with extensive contacts between Rtt105 and DBDs of RPA. While the latter indicates that Rtt105 and ssDNA compete for binding to RPA and have significant crosstalk in their RPA binding interactions, the former implies that RPA-Rtt105 interactions and RPA-ssDNA binding function independently. There is an apparent contradiction, which is not clearly explained. Is the finding that RPA has a comparable binding affinity to ssDNA with and without Rtt105 a mere coincidence?

- Our data show that the interactions between Rtt105 and the six domains (and linkers) of RPA is quite complex. The permutation of configurational changes increases as DNA binds to the complex and releases/remodels Rtt105. In both previously published papers on Rtt105-RPA-DNA interactions (Wang et. al. PNAS 2021, and Li et. al., EMBO J, 2018) a 50% enhancement in DNA binding by RPA was shown in the presence of Rtt105. There are no convenient means to reliably measure super high affinity binding of stoichiometric complexes. We employed EMSA and anisotropy (whose upper limit is $K_D \sim 1-10$ nM) to find that in presence of Rtt105, the binding affinity between RPA and DNA remains very high (and below our limits of detection). Thus, we state that any differences (if they are present) cannot be inferred from such steady-state measurements unless there is a significant reduction in binding affinity, which was not observed.

- Even in the pre-steady state experiments, while we see a difference in the rate or the amplitude of signal change, we are cognizant of the fact that the interpretations are not straightforward. If one domain were binding to a protein without cooperativity or allostery, then the rates could be used to backtrack to arrive at some understanding of binding interactions. Here, four domains are binding and dissociating, and we have a signal readout from only two domains and the end of the DNA. An ideal experiment would be to place a fluorophore on each of the six domains of RPA and observe time-dependent FRET changes as these multi-domain molecules engage with each other. We have succeeded in positioning probes on two DBDs without affecting function, but we have not been successful in probing the other domains (yet).

- Given the complexity involved in these interactions and the limit of resolution of these bulk measurements, all we can argue is that the binding affinity in presence of Rtt105 is almost as high and possible unchanged from controls (RPA + DNA alone).

2. On page 6, "RPA binds stoichiometrically to a 5'-Cy5-(dT)30 oligonucleotide in electrophoretic mobility band-shift analysis (EMSA) and a preformed Rtt105-RPA complex does not influence the ssDNA binding activity of RPA ($K_D=5.5\pm 1.1$ and 6.7 ± 2.1 nM for RPA and RPA-Rtt105, respectively; Figures 2a & b)". Based on this statement, K_D for RPA is smaller than K_D for RPA-Rtt105, but the raw data points (compare grey and blue data points) suggest an opposite trend in Figure 2b. Is this a typo?

- We performed an additional $n=2$ of these experiments and were able to get tighter error in our measurements. These are shown in Fig. 2b. Again, as explained above, we are fitting data for RPA-DNA binding that are stoichiometric and these low nM numbers are not a true reflection of

the binding affinity as the actual K_D values are likely in the pM range. Given the overlapping error bars and the uncertainty in fitting, we would like to argue that the binding isotherms \pm Rtt105 are actually quite comparable. To estimate binding isotherms of such high affinity single-strand DNA binding proteins (eg., SSB) 1-2 M NaBr can be used to weaken the complex (kindly refer to Lohman and Overman JBC 1985). However, we cannot use these conditions in the presence of Rtt105 and thus even such approaches cannot be used in this scenario.

3. Figures 2h, 3h, 6a, and 6b show a transition in FRET noise level at 1s, likely due to a change in data filtering. The authors need to explain this transition. Whenever possible, it would be more helpful to use the same filtering for a given data set.

- The stopflow data are not filtered. Data are collected as 7-10 individual traces and averaged. However, due to the rapid kinetics, data are collected in two time-windows using the SX20 software (Applied Photophysics). In the first window, 5000 points are collected over a short period 0.001 – 1 s. In the second window, 5000 points are collected from 1s – 10s. We collect more points in the first window to better capture the rapid binding/remodeling phase. The apparent transition observed at the 1s mark is observed because the first 5000 points are collected over much shorter time interval (1 second is divided into 5000 points) leading to higher density and distribution for each point. The averaged trace shows a bigger spread of data about the mean and appears noisier. In the second part, 1s-10s, 5000 points are spread over 10 seconds which leads to better s/n and thus a tighter spread. To reiterate, no separate smoothing or data-filters are applied. The raw data are presented for all stop flow experiments.

4. The story might be easier to follow if Figure 4, which shows the contact maps of Rtt105 and RPA, were presented as Figure 1.

- We tried composing the paper with the XL-data in Figure 1. However, we encounter a problem where the previously published data and models need to be first discussed before we present the new model. Thus, unfortunately, we must position the data at its current position in the manuscript.

5. The abstract places much emphasis on Rad52, but the focus on Rad52 in the main text is limited.

-Rad52 here is used as an example for RPA binding with an interacting protein. According to the suggestion, we have included more detail about Rad52 and its interactions in the Discussion section.

We thank all the reviewers again for their time and valuable comments. We hope our answers adequately address the concerns.

REVIEWERS' COMMENTS

Reviewer #1 (Remarks to the Author):

The authors have done an excellent job addressing my comments. I have no more questions.

Reviewer #2 (Remarks to the Author):

The authors have responsively addressed all my comments and the manuscript should be ready for publication. Thanks!

Reviewer #3 (Remarks to the Author):

The authors have addressed all my previous comments/suggestions. Setting the two following minor comments aside, I recommend publication of the manuscript in Nature Communications.

1. My comment regarding Fig.6 (snapping around 30s) referred to Fig.6e) with snapping at 50s - 15s (dark region) = ~ 30 s indicating the span AFTER flowing RPA and RTA. As the time span after flow in Fig.d) was 80s - 15s = 55s, I was asking for another molecule under condition in e) with a comparable time span to Fig.6d) as dissociation could occur, albeit at a slower time scale. In their rebuttal, the authors now added one new image under condition of B (similar to Fig.6d) but not C.

2. Regarding the colouring of Cy3/Cy5 in the schematics: The authors write that pink and blue are chosen as they reflect the "appearance" of the dyes. This is correct and a consequence of their absorption, but might lead to confusion as for FRET to occur, the energy is transferred from the donor dye to spectrally red (!) shifted acceptor dye. Conventionally for CY3/CY5 that would be green -> red in the fluorescence spectrum.

Reviewer #4 (Remarks to the Author):

I am satisfied with the response from the authors and recommend the publication of this manuscript in Nature Communications.

REVIEWERS' COMMENTS

Reviewer #3 (Remarks to the Author):

The authors have addressed all my previous comments/suggestions. Setting the two following minor comments aside, I recommend publication of the manuscript in Nature Communications.

1. My comment regarding Fig.6 (snapping around 30s) referred to Fig.6e) with snapping at 50s - 15s (dark region) = ~30s indicating the span AFTER flowing RPA and RTA. As the time span after flow in Fig.d) was 80s - 15s = 55s, I was asking for another molecule under condition in e) with a comparable time span to Fig.6d) as dissociation could occur, albeit at a slower time scale. In their rebuttal, the authors now added one new image under condition of B (similar to Fig.6d) but not C.

I believe this is the data for panel C that the reviewer requested. Our apologies if we missed this in our previous response statement. Data from two more independent experiments are shown for all three panels.

2. Regarding the colouring of Cy3/Cy5 in the schematics: The authors write that pink and blue are chosen as they reflect the “appearance” of the dyes. This is correct and a consequence of their absorption but might lead to confusion as for FRET to occur, the energy is transferred from the donor dye to spectrally red (!) shifted acceptor dye. Conventionally for CY3/CY5 that would be green -> red in the fluorescence spectrum.

- We changes the colors for the FRET-cartoons in all the following Figure panels: Fig 2g, 2i, 3b, 3c, 3d, 3e, 5c, 6a, 6b.

We also made the same change in all our Figures in the Supplementary Information to have this color scheme.

Reviewer #1 (Remarks to the Author):

The authors have done an excellent job addressing my comments. I have no more questions.

Reviewer #2 (Remarks to the Author):

The authors have responsively addressed all my comments and the manuscript should be ready for publication. Thanks!

Reviewer #4 (Remarks to the Author):

I am satisfied with the response from the authors and recommend the publication of this manuscript in Nature Communications.